# *All the World's a Sphere*: LEARNING EXPRESSIVE HIERARCHICAL REPRESENTATIONS WITH ISOTROPIC HYPERSPHERICAL EMBEDDINGS

## ABSTRACT

Most existing embedding frameworks rely on Euclidean geometry, which, while effective for modeling symmetric similarity, struggle to represent richer relational structures such as asymmetry, hierarchy, and transitivity. Although alternatives like hypercubes and ellipsoids introduce containment-based semantics, they often suffer from axis-aligned rigidity, anisotropic bias, and high parameter overhead. To address these limitations, we propose SpheREx (**Sphe**rical **R**epresentations for Hierarchical **Ex**pressiveness), a geometric embedding framework that utilizes isotropic hyperspheres for hierarchical and asymmetrical relation representation. By representing entities as hyperspheres, SpheREx naturally models containment, intersection, and mutual exclusion while maintaining rotational invariance and closed-form inclusion criteria. We formally characterize the geometric and probabilistic properties of hyperspherical interactions and show that they capture desirable logical structures. To ensure stable optimization and prevent uncontrolled radius growth, we introduce a volume clipping and radius regularization strategy tailored for asymmetric tasks. We conduct extensive evaluations across four diverse real-world benchmarks, spanning both text and vision modalities. SpheREx consistently outperforms 12 competitive baselines, achieving statistically significant improvements across key evaluation measures. Ablations supported by qualitative analysis across benchmarks demonstrates the efficacy of hyperspheres over state-of-the-art geometric baselines.

## 1 INTRODUCTION

Hierarchical structures are pervasive in real-world data across domains (Rashid et al., 2021). In NLP, taxonomies such as WordNet (Fellbaum et al., 1998) and SemEval (Jurgens & Pilehvar, 2016; Bordea et al., 2016) organize concepts via hypernym–hyponym relations, enabling semantic reasoning and entailment. Hierarchical structures also serve as core infrastructure in numerous high-impact applications. For instance, e-commerce platforms like Amazon (Mao et al., 2020) and Alibaba (Luo et al., 2020) use ontologies to structure product catalogs, facilitate user navigation, and personalize recommendations. Pinterest leverages hierarchical labeling for content discovery and targeted advertising (Mahabal et al., 2023). These hierarchical structures underpin a wide range of downstream services, including web content tagging (Liu et al., 2019), document retrieval (Lee et al., 2024), personalized recommendation (Huang et al., 2019), and hierarchical classification (Gao, 2020), demonstrating their critical role in structuring modern intelligent systems.

Despite the centrality of hierarchical structures in modern applications, most existing embedding methods rely on Euclidean vector spaces (Globerson et al., 2004), which are inherently limited in their ability to encode directionality, containment, and asymmetric relations as shown in Fig. 1(a). Classical text embedding methods such as Word2Vec (Mikolov et al., 2013) and BERT (Devlin et al., 2019) learn context-aware representations of textual units. While effective across many NLP tasks, these embeddings are typically situated in Euclidean space and are optimized for symmetric semantic similarity rather than relational or hierarchical structure. As shown in Fig. 1(b), they are often suboptimal for applications that require modeling of asymmetric relationships, such as hierarchical classification or entailment reasoning (Cohen-Addad et al., 2022).

Recent studies have explored geometric embeddings that extend beyond point representations to richer structures such as boxes (Li et al., 2018; Dasgupta et al., 2020; Jiang et al., 2023), enabling explicit modeling of hierarchy, transitivity, and inclusion. These methods have shown promise in capturing structural semantics through intersection and containment (c.f. Fig. 1(c)). However, axis-aligned geometric representations, such as boxes, are prone to high parameter complexity, orientation sensitivity, and local identifiability issues, wherein small changes to the parameters result in invariant model behavior, leading to ambiguous gradients. These limitations motivate the need for a representation that combines compact geometric expressiveness with smooth optimization and interpretability with isotropic relational structures.

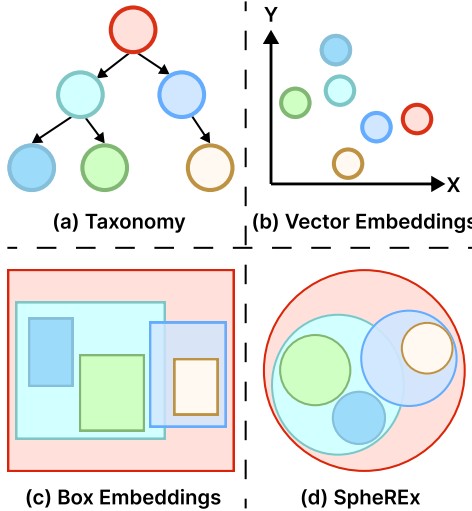

**(a) Taxonomy**  **(b) Vector Embeddings**

**(c) Box Embeddings**  **(d) SpheREx**

Figure 1: Concept representations: (a) A taxonomy with hierarchical relationships; (b) Euclidean embeddings; (c) box embeddings; (d) `SpheREx` uses isotropic hyperspheres for interpretable hierarchy and entailment.

To address these limitations, we propose `SpheREx` (**Sphe**rical **R**epresentations for Hierarchical **Ex**pressiveness), an embedding framework that models entities[1] as isotropic hyperspheres. Our formulation leverages geometric interactions, such as containment, intersection, and disjointness, to naturally encode both symmetric and asymmetric semantic relations as shown in Fig. 1(d). Isotropic hyperspheres provide closed-form inclusion criteria and exhibit rotational invariance, enabling efficient reasoning over containment without sensitivity to orientation. By enforcing an isotropic structure, `SpheREx` reduces parameter complexity and avoids local identifiability issues often encountered in high-dimensional geometrical embeddings. Furthermore, it introduces smooth inductive biases that facilitate optimization and enhance generalization. While simpler in form, hyperspherical embeddings retain the expressiveness needed to model semantic asymmetry and hierarchy through geometric containment and probabilistic overlap.[2]

To demonstrate the versatility and effectiveness of `SpheREx`, we conduct comprehensive experiments across four diverse tasks — taxonomy expansion, semantic similarity, probabilistic entailment reasoning, and image-based alignment. In taxonomy expansion tasks on the SemEval-2016 Task 13 datasets (Bordea et al., 2016), `SpheREx` significantly outperforms state-of-the-art methods, achieving substantial improvements of 21.69% in accuracy, 6.07% in mean reciprocal rank (MRR), and 5.47% in Wu & Palmer (Wu&P) similarity metrics. On the Quora Question Pairs (QQP) dataset, `SpheREx` captures symmetric semantic similarity more effectively than vector and geometric baselines, yielding the highest AUC-ROC and F1-score under various decision thresholds. For probabilistic entailment reasoning over user preferences on MovieLens-20M, `SpheREx` exhibits superior alignment with ground-truth conditional probabilities, attaining the lowest KL divergence and highest Pearson and Spearman rank correlations. Finally, in a fine-grained image similarity task on Caltech-UCSD Birds dataset (Daroya et al., 2024), `SpheREx` surpasses geometrical and probabilistic baselines, achieving improvements of 6.65% in precision, 3.99% in MRR and 12.43% in mean rank (MR). These results collectively highlight the robustness and generality of `SpheREx` framework across symmetric, asymmetric and hierarchical multimodal tasks.

## 2 RELATED WORK

Geometric embedding methods have become central for modeling hierarchical and relational semantics in language, vision and knowledge graphs (Huang et al., 2023; Xu et al., 2020). To encode hierarchy and containment, order embeddings (Vendrov et al., 2015) enforce partial ordering through

---

[1]Throughout this paper, we use the term *entities* to refer to the semantic units being embedded. Depending on the task, these include taxonomy concepts (e.g., categories in SemEval), movies (in MovieLens), questions (in Quora), or images (in Caltech-UCSD Birds).

[2]We have uploaded the source code and datasets as supplementary; we are committed to release them upon acceptance of the paper.

reversed product cones in Euclidean space, introducing asymmetry. Box embeddings (Vilnis et al., 2018; Dasgupta et al., 2020; Li et al., 2018) generalize this by modeling entities as axis-aligned hyperrectangles, enabling probabilistic reasoning through volume-based overlap. Deterministic counterparts (Jiang et al., 2023) enforce hard containment for relational edges. Despite their expressiveness, box-based models suffer from local identifiability issues, where large regions of the parameter space yield equivalent model behavior, making optimization unstable. To address these challenges, recent work has proposed smoothing the energy landscape using Gaussian convolution (Dasgupta et al., 2020), though this leads to gradient sparsity and does not resolve the alignment sensitivity. Other probabilistic set-based box embeddings, such as Query2Box (Ren et al., 2020), represent logical queries as axis-aligned boxes for reasoning over multi-hop queries in knowledge graphs, but still inherit the limitations of non-rotationally invariant geometries. Ellipsoidal embeddings (Li et al., 2022) improve flexibility over boxes by improving optimization but being anisotropic they also limit rotation.

In contrast to these anisotropic geometric methods, our approach represents each entity as an isotropic hypersphere in latent space. This removes the need for axis alignment, making our embeddings rotationally invariant and more stable during training. Unlike ellipsoids, which require more parameters and are sensitive to orientation, hyperspheres have equal radii (isotropic) in all dimensions. Our method also supports both soft and strict relational constraints, enabling accurate modeling of asymmetric and hierarchical relationships. Overall, the simplicity, efficiency, and expressiveness of hyperspheres make our method a robust alternative for structured semantic representation.

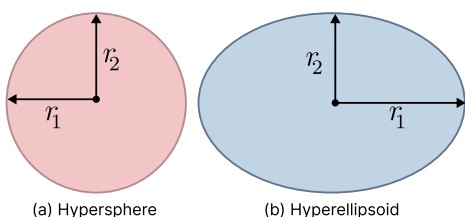

(a) Hypersphere    (b) Hyperellipsoid

Figure 2: Isotropic hypersphere ($r_1 = r_2$) vs. anisotropic ellipsoid ($r_1 \neq r_2$).

## 3 THE SPHEREx MODEL

### 3.1 HYPERSPHERE REPRESENTATION

**Definition 3.1 (Local Identifiability).** A geometric embedding space is *locally identifiable* if small changes in parameters lead to distinct semantic representations. Formally, for $\theta \in \Omega$, there exists a neighborhood $\mathcal{N}(\theta)$ such that $\mathcal{L}(x \mid \theta') \neq \mathcal{L}(x \mid \theta)$ for all $\theta' \in \mathcal{N}(\theta)$.

Box and ellipsoidal embeddings often violate this property, as different parameterizations can yield the same containment, leading to flat loss surfaces and unstable gradients (Vilnis et al., 2018; Dasgupta et al., 2020). In contrast, hypersphere embeddings improve identifiability by using fewer parameters and enforcing stricter containment. Each entity is modeled as a closed hypersphere in $\mathbb{R}^d$ as $\{x \in \mathbb{R}^d : \|x - \mathbf{c}\|_2 \leq r\}$, where $\mathbf{c} \in \mathbb{R}^d$ is the center and $r \in \mathbb{R}_+$ is the radius. Each hypersphere defines a measurable region that can be treated as a probabilistic event by assigning it an indicator variable.

**Definition 3.2 (Indicator Random Variable).** Given a probability space $(\Omega, \mathcal{E}, P)$ and event $E \in \mathcal{E}$, the indicator variable $1_E : \Omega \to \{0, 1\}$ is defined as $\int_\Omega 1_E(\omega) \, dP(\omega) = P(E)$.

**Definition 3.3 (Spherical Probability Model).** Let $(\Omega_{\text{sphere}}, \mathcal{E}, P_{\text{sphere}})$ be a probability space. A collection of probabilistic hyperspheres $\{\mathsf{Sphere}(X_i)\}_{i=1}^N$ is defined as $\mathsf{Sphere}(X_i) = \{x \in \mathbb{R}^d : \|x - \mathbf{c}_i\|_2 \leq r_i\}$ with center $\mathbf{c}_i$ and radius $r_i$. The corresponding indicator variable $X_i(x) = 1_{\mathsf{Sphere}(X_i)}(x)$ evaluates to 1 if $x$ lies inside $\mathsf{Sphere}(X_i)$. The set $\{X_1, \ldots, X_N\}$ defines a *probabilistic hypersphere model* over $(\Omega_{\text{sphere}}, \mathcal{E}, P_{\text{sphere}})$.

**Motivation for Hyperspherical Embeddings.** An $n$-dimensional hypersphere or ellipsoid $\mathcal{E} \subset \mathbb{R}^n$ is defined by a center $\mathbf{c} \in \mathbb{R}^n$ and either a scalar radius $r \in \mathbb{R}_+$ (for hyperspheres) or a positive semi-definite matrix $R \in \mathbb{R}^{n \times n}$ (for ellipsoids). While ellipsoids allow for axis-specific scaling and rotation, their anisotropic geometry introduces higher model complexity, orientation sensitivity, and increased risk of overfitting due to the larger number of parameters.

To promote geometric simplicity and reduce model capacity, we constrain representations to isotropic hyperspheres with a scalar radius. As illustrated in Fig. 2, hyperspheres (a) have equal

radii across axes ($r_1 = r_2$), unlike ellipsoids (b) with axis-specific scaling ($r_1 \neq r_2$). This isotropy yields rotational invariance and uniform curvature, enabling smoother optimization and better generalization. The uniform boundary also allows compact, principled modeling of containment, intersection, and latent hierarchies. We formalize key geometric and statistical properties below.

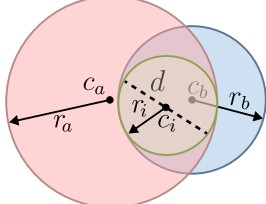

Figure 3: Modeling intersection via auxiliary hypersphere.

**Theorem 1 (Rotational Invariance).** *Let* $\mathbb{S}_r^d(\mathbf{c}) = \left\{ \mathbf{x} \in \mathbb{R}^d \mid \|\mathbf{x} - \mathbf{c}\|_2 = r \right\}$ *denote a $d$-dimensional hypersphere of radius $r > 0$ centered at $\mathbf{c} \in \mathbb{R}^d$. For any orthogonal matrix $Q \in \mathbb{R}^{d \times d}$ satisfying $Q^\top Q = I$, we have $Q \cdot \mathbb{S}_r^d(\mathbf{c}) = \mathbb{S}_r^d(Q\mathbf{c})$.*

**Theorem 2 (Uniform Hyperspherical Distribution).** *Let $\mathbf{x} \sim \mathcal{U}(\mathbb{S}^{d-1})$ be a random vector sampled uniformly from the unit hypersphere $\mathbb{S}^{d-1} \subset \mathbb{R}^d$. Then the distribution is rotationally invariant, and its density is constant over the surface. That is, for any two measurable patches $A_{\mathbf{v}_1}, A_{\mathbf{v}_2} \subseteq \mathbb{S}^{d-1}$ with equal surface measure, we have $\mathbb{P}(\mathbf{x} \in A_{\mathbf{v}_1}) = \mathbb{P}(\mathbf{x} \in A_{\mathbf{v}_2})$.*

**Theorem 3 (Compactness-Induced Generalization).** *Let $\mathcal{M}_{\text{sphere}}$ and $\mathcal{M}_{\text{ellipsoid}}$ denote the parameter manifolds of hyperspheres and ellipsoids, respectively. Then, $\dim(\mathcal{M}_{\text{sphere}}) < \dim(\mathcal{M}_{\text{ellipsoid}})$ implying that hyperspherical models exhibit lower capacity and improved generalization bounds due to reduced parameter complexity.*

The proofs of these theorems are given in Appendix A.

## 3.2 Relational Semantics in Hyperspherical Embeddings

We consider the interaction between two hyperspheres embedded in $\mathbb{R}^n$, each defined by a center $\mathbf{c}_a, \mathbf{c}_b \in \mathbb{R}^n$ and radii $r_a, r_b \in \mathbb{R}_+$, as shown in Fig. 3. The first-order relation between them is governed by the Euclidean distance, $d = \|\mathbf{c}_a - \mathbf{c}_b\|_2 = \left( \sum_{i=1}^n (c_{a,i} - c_{b,i})^2 \right)^{\frac{1}{2}}$.

Sphere interactions are determined by the distance $d$ between centers and radii $r_a, r_b$: they are disjoint if $d > r_a + r_b$; externally tangent if $d = r_a + r_b$; intersecting if $|r_a - r_b| < d < r_a + r_b$; internally tangent if $d = |r_a - r_b|$; and nested if $d < |r_a - r_b|$. These spatial configurations enable hyperspheres to represent

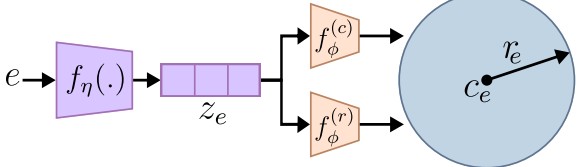

Figure 4: Projection of entity $e$ onto the hypersphere.

logical and semantic relations through geometric inclusion. We model the intersection of two hyperspheres as a hypersphere incircling the intersection, called auxiliary hypersphere. The center $\mathbf{c}_i$ and radius $r_i$ of the auxiliary hypersphere are given as $\mathbf{c}_i = \mathbf{c}_a + ((r_a - r_b + d)/(2d))(\mathbf{c}_b - \mathbf{c}_a)$ and $r_i = (r_a + r_b - d)/2$.

## 3.3 Projection to Hyperspherical Space

To encode entities in a geometry that supports structured semantic reasoning, we project each entity $e \in \mathcal{E}$ into a hyperspherical embedding space. As illustrated in Figure 4, the projection pipeline consists of two stages. First, a latent encoder $f_\eta(\cdot)$ maps the raw input $e$, such as a textual description, image, or identifier – into a dense vector representation $\mathbf{z}_e \in \mathbb{R}^d$: $\mathbf{z}_e = f_\eta(e)$. Then, a projection network $f_\phi(\cdot)$ transforms $\mathbf{z}_e$ into the parameters of a hypersphere, defined by a center $\mathbf{c}_e = f_\phi^{(c)}(\mathbf{z}_e) \in \mathbb{R}^d$ and a radius $r_e = \exp(f_\phi^{(r)}(\mathbf{z}_e)) \in \mathbb{R}_+$, where $\exp(x) = e^x$ while $f_\phi^{(c)}$ and $f_\phi^{(r)}$ are separate multilayer perceptrons (MLPs) to predict the center and radius, respectively. The $\exp(\cdot)$ function ensures non-negativity of the radius. This yields a hyperspherical embedding $\text{Sphere}(e) = (\mathbf{c}_e, r_e)$, which serves as the geometric representation of entity $e$ in latent space.

## 3.4 TRAINING OBJECTIVES

To align hyperspherical embeddings with semantic structure, we define objectives based on geometric and probabilistic principles. These operate on positive and negative pairs to capture containment, intersection, and disjointness. We jointly optimize geometric constraints, inclusion scores, and regularization terms to ensure stability and interpretability. Key objectives are defined below.

**Geometric Objectives.** To align hyperspherical embeddings with semantic relations, we define a unified margin-based objective that handles both containment and disjointness constraints. Given a pair of entities $(e_1, e_2)$, represented by hyperspheres with centers $\mathbf{c}_1, \mathbf{c}_2 \in \mathbb{R}^d$ and radii $r_1, r_2 \in \mathbb{R}_+$, we define,

$$\mathcal{L}_{\text{geom}} = \begin{cases} \max\left(0, \|\mathbf{c}_1 - \mathbf{c}_2\|_2 - (r_2 - r_1 - \gamma)\right), & \text{if } e_1 \subseteq e_2 \text{ (containment)}, \\ \max\left(0, r_1 + r_2 + \epsilon - \|\mathbf{c}_1 - \mathbf{c}_2\|_2\right), & \text{if } e_1 \perp e_2 \text{ (disjointness)}, \end{cases} \tag{1}$$

where $\gamma$ and $\epsilon$ are margin hyperparameters for containment and disjointness, respectively.

**Probabilistic Objective.** We model inclusion using a radius-based approximation of volume, inspired by prior box embedding methods. However, computing exact auxiliary hypersphere volumes is both computationally intensive and numerically unstable due to exponential scaling and numerical instability, especially when $r < 1$, leading to vanishing gradients. Fortunately, for isotropic hyperspheres, volume scales linearly with radius ($\text{Vol}(\text{Sphere}(e)) \propto r$), allowing efficient and stable approximation without explicit volume computation. We therefore use radius as a proxy for volume. Given two entities $e_1$ and $e_2$, represented as hyperspheres $\text{Sphere}(e_1)$ and $\text{Sphere}(e_2)$, we define the inclusion probability as,

$$P(e_1 \mid e_2) \approx \frac{\text{Vol}(\text{Sphere}(e_1) \cap \text{Sphere}(e_2))}{\text{Vol}(\text{Sphere}(e_2))} \approx \frac{\text{Vol}(\text{Sphere}(e_i))}{\text{Vol}(\text{Sphere}(e_2))} \approx \frac{r_i}{r_2}. \tag{2}$$

In general, we train `SpheREx` with cross-entropy losses and apply task-specific objectives depending on the nature of each task, as detailed in Section 4.

## 4 EXPERIMENTS

We conduct comprehensive experiments across diverse tasks and modalities to demonstrate the effectiveness of `SpheREx`. These include hierarchical representation, semantic entailment, recommendation, and vision-language alignment. We evaluate on multiple benchmark datasets spanning these tasks, allowing us to assess the generalizability of hyperspherical representations across symmetric, asymmetric, and hierarchical relationships across multiple modalities.

### 4.1 TAXONOMY REPRESENTATION AND EXPANSION

**Experiment Setting.** We evaluate the effectiveness of `SpheREx` on the taxonomy expansion task, representing hierarchical and asymmetrical relationships using real-world benchmark datasets – SemEval 2016 Task 13 (Bordea et al., 2016), which consists of datasets in environment (SEMEVAL16-ENV), science (SEMEVAL16-SCI) and food (SEMEVAL16-FOOD) domains. Taxonomy expansion involves adding leaf nodes to an existing taxonomy (also called "seed taxonomy"). Each dataset comprises a seed taxonomy and a set of query terms to be integrated. For each query term, we aim to identify the appropriate anchor node within the seed taxonomy to which it should be attached. For training, positive examples are constructed by pairing query terms with their correct anchor nodes, while negative examples are generated by pairing query terms with randomly selected non-parent nodes. During inference, we rank all candidate anchors in the seed taxonomy for a query node by computing the extent of probabilistic overlap with ties being broken by choosing the parent with smaller radius/volume. We employ standard evaluation metrics used in taxonomy expansion tasks, such as accuracy (Acc), Mean Reciprocal Rank (MRR), and Wu & Palmer similarity (Wu&P), to assess model performance against BERT+MLP (Devlin et al., 2019), TAXI (Panchenko et al., 2016), TaxoExpan (Shen et al., 2020), STEAM (Yu et al., 2020) and BoxTaxo (Jiang et al., 2023) baselines. The latent encoder used in this case is BERT which encodes the definitions of queries and entities in the seed taxonomy. We employ geometrical and probabilistic losses to train the model. Experimental details, including dataset statistics, baseline descriptions, preprocessing steps, loss functions, and hyperparameter settings, etc, are discussed in Appendix B.

Table 1: Performance comparison between `SpheREx` and baseline methods. Results for each method are presented as mean$^{\text{std-dev}}$ in percentage across three runs with three random seeds. The best performance is marked in bold, while the best baseline is underlined.

| Dataset | SEMEVAL16-ENV | | | SEMEVAL16-SCI | | | SEMEVAL16-FOOD | | |
|---|---|---|---|---|---|---|---|---|---|
| Metric | Acc | MRR | Wu&P | Acc | MRR | Wu&P | Acc | MRR | Wu&P |
| BERT+MLP | $12.6^{1.1}$ | $23.9^{1.6}$ | $48.3^{0.8}$ | $12.2^{1.7}$ | $19.7^{1.4}$ | $45.1^{1.1}$ | $12.7^{1.8}$ | $17.4^{1.3}$ | $49.1^{1.2}$ |
| TAXI | $18.5^{1.3}$ | N/A | $47.7^{0.4}$ | $13.8^{1.4}$ | N/A | $33.1^{0.7}$ | $20.9^{1.1}$ | N/A | $41.6^{0.2}$ |
| TaxoExpan | $10.7^{4.1}$ | $28.7^{3.8}$ | $48.5^{1.7}$ | $24.2^{5.4}$ | $40.3^{3.3}$ | $55.6^{1.9}$ | $24.6^{4.7}$ | $38.4^{3.1}$ | $52.6^{2.2}$ |
| STEAM | $\underline{34.1}^{3.4}$ | $44.3^{2.1}$ | $65.2^{1.4}$ | $\underline{34.8}^{4.5}$ | $\underline{50.7}^{2.5}$ | $\underline{72.1}^{1.7}$ | $\underline{29.5}^{5.2}$ | $39.3^{3.2}$ | $62.5^{0.8}$ |
| BoxTaxo | $32.3^{5.8}$ | $\underline{45.7}^{3.2}$ | $\underline{73.1}^{1.2}$ | $26.3^{4.5}$ | $41.1^{3.1}$ | $61.6^{1.4}$ | $28.3^{5.1}$ | $\mathbf{43.9}^{4.6}$ | $\mathbf{64.7}^{1.6}$ |
| `SpheREx` | $\mathbf{43.1}^{1.1}$ | $\mathbf{50.6}^{0.7}$ | $\mathbf{77.6}^{0.4}$ | $\mathbf{47.2}^{1.2}$ | $\mathbf{57.5}^{0.6}$ | $\mathbf{75.1}^{0.2}$ | $\mathbf{30.4}^{2.3}$ | $\underline{41.3}^{1.1}$ | $\underline{63.1}^{0.6}$ |

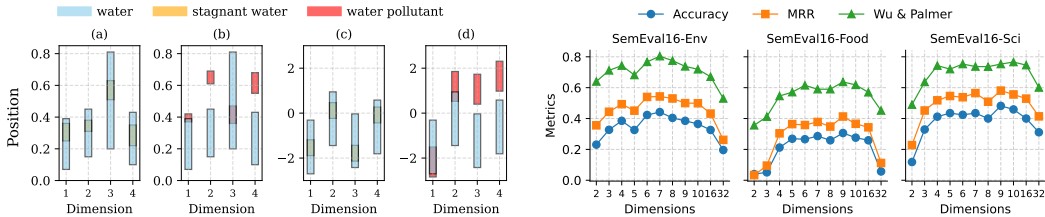

Figure 5: Comparison of boxes (a, b) and hyperspheres (c, d) for "water", "stagnant water", and "water pollutant" across four dimensions.

Figure 6: Effect of dimensionality on Acc, MRR and Wu&P on taxonomy expansion benchmarks.

**Results, Ablations and Case Study.** We observe from Table 1 that `SpheREx` consistently outperforms the baseline methods across most metrics, demonstrating a significant performance gain. However, on the SEMEVAL16-FOOD dataset, while the performance on MRR and Wu&P slightly declines due to poor quality of definitions provided, `SpheREx` still remains the strongest among all baselines on these metrics, especially accuracy. Furthermore, Fig. 5 compares the geometric behavior of box and hyperspherical embeddings. Fig. 5 (a) and (b) illustrate that axis-aligned box embeddings exhibit anisotropic behavior, with side lengths varying a lot across dimensions, increasing complexity and making boxes biased towards seen data. In contrast, Fig. 5(c) and (d) show that `SpheREx` enforces isotropy through equal-radius hyperspheres across dimensions, providing rotational invariance and more structured allocation of space. Moreover, Fig. 6 demonstrates the impact of embedding dimensionality, as performance initially improves with increasing dimensions, but degrades beyond a threshold due to the challenges of modeling in high-dimensional spaces. These results highlight the core advantages of `SpheREx`'s geometric formulation which are isotropy, space efficiency, and reduced overfitting in high-dimensional regimes. These findings underscore the importance of isotropic geometry and dimensional regularization in achieving expressive, stable, and generalizable embeddings for hierarchically structured reasoning tasks.

## 4.2 MOVIELENS

**Experiment Setting.** We also evaluate `SpheREx` on a conditional reasoning task using the MOVIELENS-20M dataset. The objective is to estimate the likelihood of a user liking movie $A$ given that they liked movie $B$, modeling asymmetric semantic dependencies common in recommendation and implication-based reasoning[3]. Following prior works (Dasgupta et al., 2020; Li et al., 2018), we extract all user-movie pairs with ratings greater than 4.0 (indicating strong preference), and prune to include only those movies with at least 4,000 user ratings. We compute the conditional probability $P(A \mid B) = \frac{\#\text{users rating both } A \text{ and } B > 4}{\#\text{users rating } B > 4}$, which serves as a weak supervision signal for annotating movie pairs. To model this relationship, we leverage the auxiliary hyperspherical intersection be-

---

[3]Implication-based reasoning infers user preferences by leveraging transitive semantic links, where liking one item (e.g., movie A) suggests a probable preference for a related item (e.g., movie B).

Table 2: Performance comparison of embedding methods with and without volume clipping. Results of Box, GumbelBox and `SpheREx` are on dimension 6. Results for each method are presented as `mean`$^{\texttt{std-dev}}$ in percentage across three runs with three random seeds. ↑ indicates higher is better, while ↓ indicates lower is better. The best performance is marked in **bold**, while the best baseline is underlined.

| Model | With Volume Clipping | | | Without Volume Clipping | | |
|---|---|---|---|---|---|---|
| | KL ↓ | Pearson ↑ | Spearman ↑ | KL ↓ | Pearson ↑ | Spearman ↑ |
| BERT | $0.0995^{0.0043}$ | $0.2414^{0.0299}$ | $0.2941^{0.0267}$ | $0.0995^{0.0043}$ | $0.2414^{0.0299}$ | $0.2941^{0.0267}$ |
| PMF | $0.6021^{0.0345}$ | $0.4891^{0.0069}$ | $0.2933^{0.0126}$ | $0.6021^{0.0345}$ | $0.4891^{0.0069}$ | $0.2933^{0.0126}$ |
| POE | $0.6222^{0.0155}$ | $0.4658^{0.0047}$ | $0.2814^{0.0072}$ | $0.6222^{0.0155}$ | $0.4658^{0.0047}$ | $0.2814^{0.0072}$ |
| Box | $0.0662^{0.0181}$ | $\underline{0.7500}^{0.0538}$ | $\underline{0.6847}^{0.0741}$ | $0.0711^{0.0232}$ | $0.7508^{0.0550}$ | $\underline{0.6869}^{0.0774}$ |
| GumbelBox | $\underline{0.0401}^{0.0133}$ | $0.7195^{0.0233}$ | $0.6519^{0.0287}$ | $\mathbf{0.0344}^{0.0042}$ | $\underline{0.7570}^{0.0678}$ | $0.6839^{0.0832}$ |
| `SpheREx` | $\mathbf{0.0397}^{0.0073}$ | $\mathbf{0.8118}^{0.0328}$ | $\mathbf{0.7897}^{0.0514}$ | $\underline{0.0400}^{0.0079}$ | $\mathbf{0.8118}^{0.0328}$ | $\mathbf{0.7902}^{0.0523}$ |

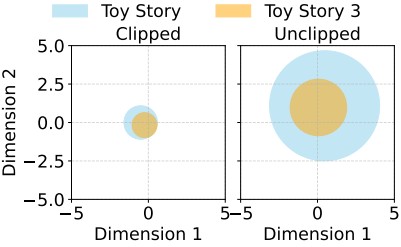

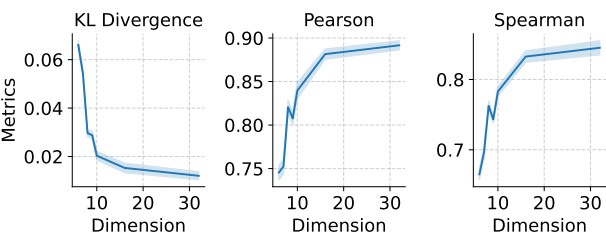

Figure 7: Example of clipped and un-clipped volumes.

Figure 8: Effect of increasing embedding dimensionality on model performance for the MovieLens-20M dataset.

tween embedding regions and frame the task as a regression problem, where the model is trained on these weak probabilistic labels to predict the conditional probability $P(A \mid B)$. We learn the conditional probability by optimizing the model using mean square error (MSE) loss. We evaluate model performance on the test set using KL divergence, Pearson, and Spearman rank correlations between the predicted and empirical probabilities. The latent encoder utilized here is BERT.

**Results, Ablations and Case Study.** As shown in Table 2, `SpheREx` achieves the best performance over baselines – BERT (Devlin et al., 2019), PMF (Mnih & Salakhutdinov, 2007), POE (Lai & Hockenmaier, 2017), Box (Dasgupta et al., 2020) and GumbelBox (Li et al., 2018) across KL divergence, Spearman and Pearson rank correlations. We compare two architectural variants of `SpheREx` – one with unconstrained hypersphere volumes and another with clipped volume constraints, as illustrated in Fig. 7 for movies "Toy Story" and "Toy Story 3". While unconstrained volume expansion is beneficial in hierarchical settings such as taxonomy expansion, where accommodating nested hyperspheres is advantageous, it introduces challenges in asymmetric modeling tasks. Specifically, unlike box embeddings that can leverage anisotropy to enforce disjointness along selective dimensions, isotropic hyperspheres require uniformly larger radii to achieve separation, which can lead to excessive capacity and diminished discrimination amongst entities. We find that applying volume clipping effectively regularizes this behavior, leading to improved performance in asymmetric reasoning. For baselines not based on geometric embeddings, such as BERT, POE, and PMF, we use the same results in both cases. Moreover, Fig. 8 shows that the performance improves on increasing dimensions for asymmetrical relationship loss. Details regarding the inference setup, dataset statistics, preprocessing steps, loss formulations, and hyperparameter settings, are provided in Appendix C.

### 4.3 DOCUMENT SIMILARITY ON QUORA QUESTION PAIRS

**Experiment Setup.** We analyze the symmetric property modeling capability of isotropic hyperspheres using the Quora Question Pairs (QQP) dataset, which involves determining whether two questions express the same meaning. This task requires capturing symmetric semantic similarity

Table 3: Performance comparison of embedding methods under different decision thresholds (25%, 50%, 75%). The best performance is marked in **bold**, while the best baseline is underlined.

| Threshold | 25% | | | 50% | | | 75% | | | |
|-----------|-----------|--------|--------|-----------|--------|--------|-----------|--------|--------|---------|
| Model | Precision | Recall | F1 | Precision | Recall | F1 | Precision | Recall | F1 | AUC-ROC |
| BERT | **0.6585** | 0.7559 | 0.7038 | 0.6816 | 0.7325 | 0.7061 | 0.7041 | 0.7056 | 0.7048 | 0.8359 |
| POE | 0.6398 | 0.7180 | 0.6785 | 0.6583 | 0.7120 | 0.6827 | 0.6820 | 0.6845 | 0.6831 | 0.8034 |
| Box | 0.5967 | 0.9386 | 0.7296 | 0.6782 | **0.8751** | 0.7642 | 0.8006 | 0.7085 | **0.7517** | 0.8960 |
| GumbelBox | 0.4251 | **0.9942** | 0.5956 | 0.6600 | 0.8011 | 0.7238 | 0.6151 | 0.8622 | 0.7180 | 0.8560 |
| SpheREx | 0.6378 | 0.9432 | **0.7604** | **0.7433** | 0.8568 | **0.7873** | **0.8705** | 0.5347 | 0.6625 | **0.9099** |

Table 4: Performance comparison on the Caltech-UCSD Birds-200-2011 dataset. ↑ indicates higher is better, while ↓ indicate lower is better. Results for each method are presented as $\text{mean}^{\text{std-dev}}$ in percentage across three runs with three random seeds. The best performance is marked in **bold**, while the best baseline is underlined.

| Model | Without Volume Clipping | | | With Volume Clipping | | |
|-------|-----------|------|-------|-----------|------|-------|
| | Precision ↑ | MR ↓ | MRR ↑ | Precision ↑ | MR ↓ | MRR ↑ |
| $\text{CLIP}_1$ | $0.0513 \pm 0.004$ | $10.568 \pm 1.612$ | $0.1750 \pm 0.625$ | $0.0513 \pm 0.004$ | $10.568 \pm 1.612$ | $0.1750 \pm 0.625$ |
| $\text{CLIP}_2$ | $0.1154 \pm 0.051$ | $8.241 \pm 1.131$ | $0.3166 \pm 0.241$ | $0.1154 \pm 0.051$ | $8.241 \pm 1.131$ | $0.3166 \pm 0.241$ |
| Box | $0.7354 \pm 0.024$ | $1.432 \pm 0.57$ | $0.8479 \pm 0.034$ | $0.8119 \pm 0.009$ | $1.607 \pm 0.26$ | $0.8823 \pm 0.012$ |
| GumbelBox | $0.7393 \pm 0.035$ | $1.513 \pm 0.32$ | $0.8487 \pm 0.011$ | $0.8974 \pm 0.061$ | $1.201 \pm 0.46$ | $0.9397 \pm 0.017$ |
| SpheREx | $\mathbf{0.7820 \pm 0.021}$ | $\mathbf{1.269 \pm 0.34}$ | $\mathbf{0.8835 \pm 0.019}$ | $\mathbf{0.9572 \pm 0.014}$ | $\mathbf{1.051 \pm 0.25}$ | $\mathbf{0.9772 \pm 0.015}$ |

between questions. The QQP dataset consists of question pairs labeled as duplicates (positive) or non-duplicates (negative). We represent each question as a hypersphere and enforce that positive pairs have overlapping hyperspheres with similar volumes, while negative pairs remain disjoint. Specifically, we compute conditional inclusion probabilities $P(A \mid B)$ and $P(B \mid A)$ to ensure symmetric relationship and avoid hierarchical correlation. A pair of questions is predicted as negative (not similar) if their corresponding hyperspheres are disjoint, and positive otherwise. However, disjointness alone can be insufficient, as even a small overlap would yield a positive prediction. Therefore, we introduce a threshold-based criterion wherein if the overlap between the two hyperspheres falls below a specified ratio, the pair is classified as negative or disjoint. The evaluation metrics used are Precision, Recall, F1-Score, and AUC-ROC. The latent encoder used in this case is BERT which encodes the question pairs. Details on how the overlaps are computed are provided in Appendix D.

**Results and Ablations.** SpheREx achieves the highest AUC-ROC score, outperforming all baseline methods, namely BERT (Devlin et al., 2019), Box (Dasgupta et al., 2020), POE (Lai & Hockenmaier, 2017) and GumbelBox (Li et al., 2018) as shown in Table 3. This indicates the strong ability of SpheREx to capture symmetric semantic similarity between question pairs. Fig. 9 shows that at a 50% overlap threshold, the model maintains a good balance between Precision and F1-score, achieving highest F1 score, although as threshold increases, the precision also increases. Therefore, at 50% threshold, SpheREx yields optimal performance. Full details of the experimental setup, including dataset statistics, baseline configurations, metric computation, loss functions, and hyperparameter settings, are discussed in Appendix D.

## 4.4 IMAGE SIMILARITY

**Experiment Setting.** We assess SpheREx's capability to capture asymmetric semantic relationships in the visual modality using the Caltech-UCSD Birds-200-2011 dataset, adhering to the protocol established in (Daroya et al., 2024). We subsample 20 semantic classes from the full dataset to construct training and evaluation subsets. The task involves aligning fine-grained visual instances (bird images) with their corresponding class-level semantic representations. Training is performed on triplets comprising an image, its associated ground-truth class, and a randomly sampled negative class. Hyperspherical embeddings are optimized such that the image hypersphere is fully contained

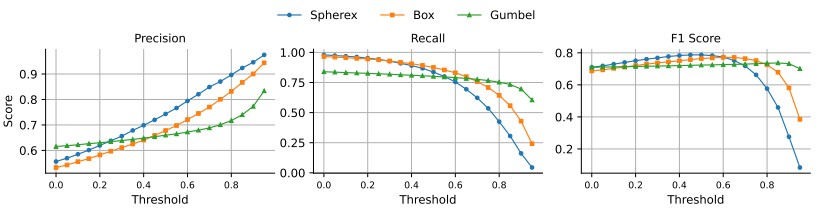

Table 5: *p*-values across datasets.

| Dataset | p-value |
|---------|---------|
| MovieLens | 0.0217 |
| Birds | 0.0172 |
| Quora | 0.0474 |
| Taxonomy | 0.0427 |

Figure 9: Performance comparison across varying thresholds on QQP dataset.

within its true class hypersphere while remaining disjoint from the negative class hypersphere. Additionally, inter-class label hyperspheres are constrained to remain disjoint using margin-based disjoint loss to enforce semantic exclusiveness across class representations. Both image and label embeddings are generated using the CLIP ViT-H/14 encoder pretrained on the LAION-2B dataset. We compare against two CLIP-based baselines – $CLIP_1$, which concatenates latent representations of image and label followed by a binary classifier trained with logistic loss; and $CLIP_2$, which employs a margin-based ranking loss to optimize semantic alignment between image and label embeddings. We evaluate the performance of SpheREx against baselines using Precision, Mean Rank (MR) and MRR given that it is an asymmetrical labeling task requiring ranking of labels. Details such as data processing, model training, inference, evaluation metrics are provided in Appendix E.

**Results and Case Study.** Table 4 shows that SpheREx consistently outperforms all baselines, demonstrating its effectiveness in modeling multimodal semantic relationships in a shared space. SpheREx achieves improvements of 6% in precision, 25% in MR, and 4.5% in MRR without volume clipping, and 6.65%, 12.43%, and 3.99% improvements, respectively, with clipping. Among the baselines, Box and GumbelBox perform best, underscoring the utility of geometric embeddings for capturing asymmetry in multimodal reasoning tasks. Furthermore, the consistent gains observed across all models after applying volume clipping highlight the importance of constraining representational capacity to preserve disjointness and containment in asymmetrical settings.

## 4.5 STATISTICAL TESTS

To assess the statistical significance of performance improvements achieved by SpheREx, we conduct paired t-tests against the strongest baseline for each dataset. As shown in Table 5, all *p*-values fall below the standard significance threshold of 0.05. This confirms that the observed gains are not due to random chance and are statistically significant across all evaluated domains. Specifically, we observe that the gains are most pronounced on recommendation and image-classification tasks, and smaller (but still significant) on taxonomy expansion and document similarity. This pattern suggests that SpheREx is particularly effective for multimodal information-retrieval/semantic alignment settings while remaining competitive on purely textual tasks. Overall, these results underscore the robustness and cross-domain consistency of SpheREx's performance.

## 5 CONCLUSION

We proposed SpheREx, a unified hyperspherical embedding framework that models entities as isotropic hyperspheres to encode both symmetric and asymmetric semantic relationships through geometric interactions like containment, intersection, and disjointness. Our formulation provided closed-form inclusion criteria, compact parameterization, and rotational invariance, enabling interpretable and efficient reasoning in both textual and visual modalities. We conducted extensive evaluations across four diverse tasks – taxonomy expansion on SemEval datasets, probabilistic reasoning on MovieLens, semantic similarity on Quora, and fine-grained visual alignment on Caltech-UCSD Birds-200-2011. SpheREx consistently outperformed strong baselines, including vector-based, probabilistic, and box-based models. Specifically, it achieved up to a 10% accuracy and 5.38% MRR gain over BoxTaxo in taxonomy expansion, reduces KL divergence by up to 25% on MovieLens compared to Box and GumbelBox, and achieves superior F1 and AUC-ROC scores over BERT and POE on Quora. Furthermore, SpheREx outperformed classical and geometric vision-language baselines in multimodal alignment tasks. These results validate the expressiveness and robustness of hyperspherical embeddings for structured representation learning across modalities.

## REPRODUCIBILITY STATEMENT

We release code, configuration files, and scripts to reproduce all results, figures, and tables. The package includes data–preprocessing code with exact splits, training and evaluation scripts, and metric computation. We provide pretrained checkpoints and fixed random seeds, and we log software/hardware versions; deterministic flags are enabled where possible. Minor nondeterminism due to inherent computational randomness may remain but does not affect conclusions. A single A100 80 GB GPU is sufficient; see Appendices B,C, D and E.

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

APPENDIX

# A  PROOF OF THEOREMS

## A.1  PROOF OF THEOREM 1 - ROTATIONAL INVARIANCE

*Proof.* Let $\mathbf{x} \in \mathbb{S}_r^d(\mathbf{c})$, so that $\|\mathbf{x} - \mathbf{c}\|_2 = r$. Consider the transformed point $\mathbf{z} = Q\mathbf{x}$. Then:

$$\|\mathbf{z} - Q\mathbf{c}\|_2 = \|Q\mathbf{x} - Q\mathbf{c}\|_2 = \|Q(\mathbf{x} - \mathbf{c})\|_2.$$

Since $Q$ is orthogonal, it preserves the Euclidean norm:

$$\|Q(\mathbf{x} - \mathbf{c})\|_2 = \|\mathbf{x} - \mathbf{c}\|_2 = r.$$

Thus, $\mathbf{z} \in \mathbb{S}_r^d(Q\mathbf{c})$, proving:

$$Q \cdot \mathbb{S}_r^d(\mathbf{c}) \subseteq \mathbb{S}_r^d(Q\mathbf{c}).$$

To prove the reverse inclusion, let $\mathbf{z} \in \mathbb{S}_r^d(Q\mathbf{c})$. By definition of the hypersphere, this implies:

$$\|\mathbf{z} - Q\mathbf{c}\|_2 = r.$$

Define $\mathbf{x} = Q^\top \mathbf{z}$. Since $Q^\top$ is also orthogonal (being the inverse of $Q$), we have:

$$\|\mathbf{x} - \mathbf{c}\|_2 = \|Q^\top \mathbf{z} - \mathbf{c}\|_2 = \|Q^\top(\mathbf{z} - Q\mathbf{c})\|_2 = \|\mathbf{z} - Q\mathbf{c}\|_2 = r.$$

So $\mathbf{x} \in \mathbb{S}_r^d(\mathbf{c})$, and since $\mathbf{z} = Q\mathbf{x}$, we get $\mathbf{z} \in Q \cdot \mathbb{S}_r^d(\mathbf{c})$. This proves:

$$\mathbb{S}_r^d(Q\mathbf{c}) \subseteq Q \cdot \mathbb{S}_r^d(\mathbf{c}).$$

Combining both inclusions, we conclude:

$$Q \cdot \mathbb{S}_r^d(\mathbf{c}) = \mathbb{S}_r^d(Q\mathbf{c}).$$

$\square$

## A.2  PROOF OF THEOREM 2 - UNIFORM HYPERSPHERICAL DISTRIBUTION

*Proof.* Let $\mathbf{x} \sim \mathcal{U}(\mathbb{S}^{d-1})$ denote the uniform distribution on the unit hypersphere $\mathbb{S}^{d-1} \subset \mathbb{R}^d$. By definition of the uniform distribution on a continuous manifold, the probability density function $f(\mathbf{x})$ is constant with respect to the surface (Hausdorff) measure on $\mathbb{S}^{d-1}$:

$$f(\mathbf{x}) = \frac{1}{|\mathbb{S}^{d-1}|}, \quad \forall \mathbf{x} \in \mathbb{S}^{d-1}.$$

where $|\mathbb{S}^{d-1}|$ denotes the surface area of the hypersphere.

Now consider any two measurable subsets $A_1, A_2 \subseteq \mathbb{S}^{d-1}$ such that $\mu(A_1) = \mu(A_2)$, where $\mu$ is the surface measure. Let $Q \in \mathrm{SO}(d)$ be an orthogonal transformation (rotation) such that $Q(A_1) = A_2$. Because the uniform distribution on $\mathbb{S}^{d-1}$ is invariant under rotations, we have:

$$\mathbb{P}(\mathbf{x} \in A_1) = \mathbb{P}(\mathbf{x} \in QA_1) = \mathbb{P}(\mathbf{x} \in A_2).$$

Thus, any two sets with equal surface area have equal probability mass under the uniform distribution. In particular, for any points $\mathbf{v}_1, \mathbf{v}_2 \in \mathbb{S}^{d-1}$ and any neighborhoods $A_{\mathbf{v}_1}, A_{\mathbf{v}_2}$ around them with equal measure, we have:

$$\mathbb{P}(\mathbf{x} \in A_{\mathbf{v}_1}) = \mathbb{P}(\mathbf{x} \in A_{\mathbf{v}_2}).$$

This proves that the density is symmetric and rotationally invariant across the sphere. $\square$

## A.3  PROOF OF THEOREM 3 - COMPACTNESS-INDUCED GENERALIZATION

*Proof.* We compare the intrinsic dimensionality of the parameter manifolds underlying hyperspherical and hyperellipsoidal representations in $\mathbb{R}^d$, which serve as proxies for model capacity in geometric embedding spaces.

**(1) Hyperspherical Manifold:** A hypersphere in $\mathbb{R}^d$ is uniquely specified by:

- A center vector $\mathbf{c} \in \mathbb{R}^d$ (translation),

- A scalar radius $r \in \mathbb{R}_+$ (isotropic scaling).

The parameter space of all hyperspheres thus forms a manifold of dimension:

$$\dim(\mathcal{M}_{\text{sphere}}) = \dim(\mathbf{c}) + \dim(r) = d + 1.$$

**(2) Hyperellipsoidal Manifold:** A hyperellipsoid is characterized by:

- A center vector $\mathbf{c} \in \mathbb{R}^d$,

- A symmetric positive semi-definite matrix $A \in \mathbb{R}^{d \times d}$ encoding axis-specific scaling and rotation.

The space of symmetric $d \times d$ matrices is of dimension:

$$\dim(A) = \frac{d(d+1)}{2}.$$

Therefore, the total number of degrees of freedom is:

$$\dim(\mathcal{M}_{\text{ellipsoid}}) = d + \frac{d(d+1)}{2}.$$

**(3) Dimensional Comparison:** Subtracting yields:

$$\dim(\mathcal{M}_{\text{ellipsoid}}) - \dim(\mathcal{M}_{\text{sphere}}) = \frac{d(d+1)}{2} - 1 > 0 \quad \text{for all } d \geq 2.$$

This shows that the hyperellipsoidal manifold admits strictly more geometric degrees of freedom than the hyperspherical manifold.

**(4) Implication for Generalization:** From a statistical learning perspective, models defined on higher-dimensional parameter spaces have increased capacity, which can lead to overfitting without sufficient regularization. The compactness and isotropy of hyperspherical embeddings induce stronger inductive biases and restrict the hypothesis class, which typically improves generalization bounds (e.g., via lower Rademacher complexity or VC-dimension).

Hence, the dimensional compactness of $\mathcal{M}_{\text{sphere}}$ relative to $\mathcal{M}_{\text{ellipsoid}}$ implies tighter generalization guarantees under standard complexity measures in statistical learning theory. $\qquad\square$

# B TAXONOMY EXPANSION

As discussed in Experiments (Section 4), taxonomy expansion task is addition of new entities to an existing taxonomy. We discuss the preliminaries of the task, data generation, model training, inference and evaluation.

**Preliminaries.** We discuss the notations, formal problem statement of taxonomy expansion as,

**Definition B.1. Taxonomy:** A taxonomy $\mathcal{T}^o = (\mathcal{N}^o, \mathcal{E}^o)$ is defined as a hierarchical directed acyclic graph (DAG), where each node $n \in \mathcal{N}^o$ denotes a concept, and each directed edge $\langle n_p, n_c \rangle \in \mathcal{E}^o$ captures a semantic "parent-to-child" relationship from parent node $n_p$ to its child $n_c$.

A taxonomy $\mathcal{T}^o$, commonly referred to as a seed taxonomy, is often constructed manually and therefore tends to be small in scale and lacking in coverage. As new concepts continuously arise, a key challenge is to accurately incorporate these unseen entities into the existing structure of $\mathcal{T}^o$. In this work, we formally define this challenge as the taxonomy expansion problem.

**Definition B.2. Taxonomy Expansion:** Given a seed taxonomy $\mathcal{T}^o = (\mathcal{N}^o, \mathcal{E}^o)$ and a set of emerging concepts $C$, the task is to update the seed taxonomy to $\mathcal{T} = (\mathcal{N}^o \cup \mathcal{C}, \mathcal{E}^o \cup \mathcal{R})$, where $\mathcal{R}$ is the set of newly created relationships linking existing entities $\mathcal{E}^o$ with emerging entities $C$. Since surface names of entities alone lack true semantics, entity descriptions $D$ are used to augment representations. Moreover, during inference, query node $q \in \mathcal{C}$ identifies its best-suited parent node $n_p \in \mathcal{N}^o$ by maximizing the matching score ($n_p = \arg\max_{a \in \mathcal{N}^o} f(a, q)$).

**Self-supervised Data Generation.** Given an initial taxonomy $\mathcal{T}^o = (\mathcal{N}^o, \mathcal{E}^o)$, we construct training data via a self-supervised strategy. Specifically, we begin by withholding 20% of the leaf nodes to serve as a test set, while the remaining portion of the taxonomy is retained as the training taxonomy. Within this training structure, each edge $\langle n_p, n_c \rangle \in \mathcal{E}^o$–where $n_p$ denotes the parent and $n_c$ the child or query node–forms a positive training pair $\langle n_p, n_c \rangle$. To obtain negative examples, we fix $n_c$ as the query term and randomly draw $N$ anchor nodes $\{n'_{p_l}\}_{l=1}^N$ from the training taxonomy. These anchors are structurally related to $n_c$ through relations like siblings, cousins, uncles, or ancestors, while explicitly excluding its true parent or any of its descendants. For each edge, we thus form a training instance $\mathbf{X}$ consisting of one positive and $N$ negative pairs, $\mathbf{X} = \{\langle n_p, n_c \rangle, \langle n'_{p_1}, n_c \rangle, \ldots, \langle n'_{p_N}, n_c \rangle\}$. By iterating over all edges in $\mathcal{E}^o$, we compile the complete self-supervised training dataset: $\mathbb{X} = \{\mathbf{X}_1, \ldots, \mathbf{X}_{|\mathcal{E}^o|}\}$.

**Model Training.** We optimize our model using a weighted combination of geometric and probabilistic loss functions, each enforcing distinct structural and semantic constraints as described in Section 3.4. We use Mean Squared Error loss for geometric containment and disjoint. These loss functions are designed to penalize undesired intersections between incompatible hyperspherical embeddings, particularly for negative samples. However, a trivial solution where an embedding collapses to an infinitesimally small volume would artificially minimize both losses without learning meaningful representations. To mitigate this, we introduce a radius regularization term that enforces a lower bound on the radius of each entity hypersphere. Specifically, for an entity embedding $B_e$ with radius $r_e$, we define the regularization loss as:

$$L_r = (r_e - \phi)^2,$$

where $\phi$ is a predefined threshold ensuring that $r_e$ remains sufficiently large to preserve the expressiveness of the learned hyperspherical representation.

To help in containment of child hypersphere inside the parent hypersphere, we aim to bring the child closer to the parent than other possible candidates. We implement a contrastive loss on the distance between the centers of the nodes. Let $d^+$ denote the distance between the center of the hypersphere corresponding to the child and the center of hypersphere of its corresponding parent and $d^-$ denote the distance of center of the child hypersphere to the center of hypersphere of negative parent. The contrastive loss for a single training sample is defined as:

$$L_{con} = \max(0, d^+ - d^-).$$

For large taxonomies, nodes at greater depths may be forced to small volumes, introducing a bias when being considered as candidates for query nodes. To enforce geometric consistency between nested hyperspheres, we introduce a penalty based on the radius ratio between a child and its parent hypersphere. Let $r_c$ and $r_p$ denote the child and parent radii, and $d$ be the dimensionality of the space. The ratio of volumes is given by:

$$v = \left( \frac{r_c}{r_p} \right).$$

We apply a soft penalty when $v < \rho_{\min}$, using a smooth weight defined as:

$$L_{rad} = (\sigma(\lambda(\rho_{\min} - v)))^2,$$

where $\sigma(\cdot)$ is the sigmoid function and $\lambda$ is a sharpness hyperparameter.

The overall training objective is defined as a weighted combination of distinct loss components:

$$L_{\text{total}} = \alpha L_{\text{geom}} + \gamma(L_r + L_{\text{con}} + L_{\text{rad}}) + \delta L_{\text{prob}},$$

where $L_{\text{geom}}$ enforces geometric constraints such as containment and disjointness, $L_{\text{prob}}$ aligns predicted and target probabilities via probabilistic scoring, $L_r$ regularizes the minimum radius, $L_{\text{con}}$ penalizes undesired containment violations, and $L_{\text{rad}}$ enforces smooth radius scaling. The coefficients $\alpha$, $\gamma$, and $\delta$ modulate the relative importance of each component.

**Inference.** Given a query concept $c \in \mathcal{C}$, the objective during inference is to identify its most semantically appropriate parent $n_p \in \mathcal{N}^o$ from the existing seed taxonomy. For each candidate $n_p$, we compute a *probabilistic containment score* using the normalized volume of the auxiliary

Table 6: Hyperparameters for Taxonomy Expansion

| Name | Value | Significance |
|---|---|---|
| n | 3 | number of layers in hypersphere projector |
| hidden | 64 | Dimension of hidden layer in hypersphere projector |
| dropout | 0.05 | Dropout rate for regularization |
| margin | 0.05 | Minimum margin enforced for containment |
| $\epsilon$ | -0.1 | Minimum margin enforced for disjointness |
| $\phi$ | 0.5 | Minimum radius of hypersphere |
| $\alpha$ | 0.25 | Weight of geometric loss |
| $\gamma$ | 1.0 | Weight of regularization loss |
| $\delta$ | 0.5 | Weight of probabilistic loss |
| negsamples | 25 or 50 | Number of negative parent samples |
| $\rho_{min}$ | 0.2 | Minimum ratio of $r_{\text{child}}/r_{\text{parent}}$ |
| $lambda$ | 8 | Sharpness parameter for sigmoid function in radius ratio loss |
| epochs | 30 | Number of training epochs |
| batch_size | 200 | Batch size |
| lr | $2 \times 10^{-5}$ | Learning rate |
| $\theta$ | 0.3 | Weight of distance metric used in final scoring |
| Optimizer | AdamW | Optimizer used |
| Seeds | 42, 97, 137 | Seeds used |

hypersphere formed by the intersection of the embeddings of $c$ and $n_p$. In parallel, we evaluate the Euclidean distance between the centers of the hyperspherical embeddings of $c$ and $n_p$, normalize these distances across all candidates, and invert them to yield a similarity-based distance score. The final matching score is computed as a convex combination of the containment and distance-based similarity scores. Candidates are ranked based on these composite scores, and the top-scoring node is selected as the predicted parent. Although the current framework selects a single parent (top-1), it can be naturally extended to a top-$k$ setting for multi-parent or hierarchical candidate expansion.

**Implementation Details.** SpheREx is implemented using PyTorch, with the baselines, excluding BERT+MLP, sourced from the respective repositories of their original authors. We finetune all parameters of the latent projector model BERT for SpheREx. All training and inference tasks were conducted on an 48GB NVIDIA A6000 GPU to ensure high computational efficiency. Hyperparameters are discussed in Table 6.

**Evaluation Metrics.** During inference, both the baselines and SpheREx rank all candidate terms for each query node. For a given the query set $\mathcal{C}$, the predictions generated by baselines and SpheREx are represented as $\{\hat{y}_1, \hat{y}_2, \cdots, \hat{y}_{|\mathcal{C}|}\}$ while the corresponding true labels are represented as $\{y_1, y_2, \cdots, y_{|\mathcal{C}|}\}$. Following BoxTaxo (Jiang et al., 2023), we adopt three metrics to evaluate the performance of baselines and SpheREx as follows,

- **Accuracy (Acc):** It counts the number of predicted parent for each query term exactly matching the ground-truth parent as,

$$\text{Acc} = \text{Hit@1} = \frac{1}{|\mathcal{C}|} \sum_{i=1}^{|\mathcal{C}|} \mathbb{I}(y_i = \hat{y}_i), \tag{3}$$

  where $\mathbb{I}(\cdot)$ represents the indicator function.

- **Mean Reciprocal Rank (MRR):** It computes the average reciprocal rank of the query term's true hypernym among within the predicted candidate list as,

$$\text{MRR} = \frac{1}{|\mathcal{C}|} \sum_{i=1}^{|\mathcal{C}|} \frac{1}{\text{rank}(y_i)}. \tag{4}$$

- **Wu & Palmer Similarity (Wu&P)**: It measures the closeness of the predicted term with the ground-truth parent based on their depth and the depth of their least common ancestor (LCA) in

Table 7: Hyperparameters for experiments on MovieLens

| Name | Value | Significance |
|---|---|---|
| n | 3 | number of layers in hypersphere projector |
| hidden | 64 | Dimension of hidden layer in hypersphere projector |
| embed_size | 6 | Dimensionality of the embedding space |
| batch_size | 512 | Batch size |
| lr | $2 \times 10^{-5}$ | Learning rate |
| $\delta$ | 1.0 | Weight of probabilistic loss |
| $\gamma$ | 1.0 | Weight of regularization loss |
| $\phi$ | 0.05 | Minimum radius of hyperspheres |
| epochs | 60 | Number of training epochs |
| Optimizer | AdamW | Optimizer used |
| Seeds | 42, 97, 137 | Seeds used |

the taxonomy as,

$$\text{Wu\&P} = \frac{1}{|\mathcal{C}|} \sum_{i=1}^{|\mathcal{C}|} \frac{2 \times \text{DEPTH}\left(\text{LCA}\left(\hat{y}_i, y_i\right)\right)}{\text{DEPTH}\left(\hat{y}_i\right) + \text{DEPTH}\left(y_i\right)}, \tag{5}$$

where $\text{DEPTH}(\cdot)$ is the depth of a node in the seed taxonomy.

## C    MOVIELENS

As discussed in Experiments (Section 4), We construct an item-item interaction dataset from user-item ratings. We discuss the data processing, metrics used, model training, and evaluation.

**Data Preprocessing.**    We construct a filtered subset of the MovieLens-20M dataset by retaining only those user-item interactions where the rating is greater than 4, thereby focusing on strong positive preferences. To ensure sufficient statistical support, we restrict the movie set to those with at least 4000 ratings, resulting in a total of 418 unique movies. We then perform a stratified 90:10 split over these movies to create disjoint training and test sets. The training set comprises ordered movie pairs $(M_a, M_c)$ used to estimate conditional probabilities $P(M_a \mid M_c)$, where $M_a \in A$ denotes a candidate antecedent and $M_c \in C$ denotes a conditioning context from the training set $C \subset A$. This yields a total of 156,793 training samples. For each movie, we construct a textual context by concatenating its title and genre descriptors, which is then tokenized and encoded using a pretrained BERT encoder. The resulting [CLS] token embedding is used as the input representation for the model.

**Model Training.**    For each training pair $(M_1, M_2)$, we SpheREx predicts conditional probability $P_{\text{pred}}(M_1 \mid M_2)$ by evaluating the normalized volume of intersection between the hyperspherical embeddings of the two movies using auxiliary hypersphere. This probabilistic containment score is regressed against the empirical ground-truth estimate $P_{\text{true}}(M_1 \mid M_2)$ using the Mean Squared Error (MSE) loss $L_{\text{mse}} = (P_{\text{true}} - P_{\text{pred}})^2$. To prevent hyperspheres from collapsing or growing excessively, thereby trivializing the containment computation, we apply a radius regularization penalty to each entity embedding.

**Evaluation.**    For evaluation, we treat each held-out movie $M_b \in B = A \setminus C$ as the query item and estimate the conditional probabilities $P(M_a \mid M_b)$ for all $M_a \in A \setminus \{M_b\}$, resulting in a total of 17,557 test instances. For each query movie $M_b$, we construct the ground-truth probability vector $\mathbf{P}$ based on empirical co-occurrence statistics from user ratings, and the corresponding predicted vector $\hat{\mathbf{P}}$ generated by the model. Both vectors are normalized to form valid probability distributions. Evaluation metrics, KL divergence, Pearson correlation, and Spearman rank correlation, are computed between $\mathbf{P}$ and $\hat{\mathbf{P}}$ for each query. The final performance is reported by averaging these metric values across all test movies $M_b \in B$.

**Evaluation Metrics.** To evaluate the performance of our model and baselines on the probabilistic reasoning task using the MovieLens-20M dataset, we employ three standard metrics that assess the alignment between the predicted and ground-truth conditional probabilities: KL divergence, Pearson correlation, and Spearman rank correlation. Let the predicted conditional probability distribution over movie pairs be denoted as $\{\hat{P}_i\}_{i=1}^{|\mathcal{C}|}$ and the corresponding empirical probabilities derived from user data as $\{P_i\}_{i=1}^{|\mathcal{C}|}$, where $\mathcal{C}$ denotes the set of movie pairs:

- **Kullback–Leibler Divergence (KL):** It quantifies the divergence between the predicted distribution $\hat{P}_i$ and the ground-truth distribution $P_i$ as,

$$\text{KL} = \frac{1}{|\mathcal{C}|} \sum_{i=1}^{|\mathcal{C}|} P_i \log \left( \frac{P_i}{\hat{P}_i} \right), \tag{6}$$

where lower KL indicates better alignment with the empirical distribution.

- **Pearson Correlation Coefficient:** It measures the linear correlation between the predicted and true probabilities,

$$\text{Pearson} = \frac{\text{Cov}(\hat{P}, P)}{\sigma_{\hat{P}} \cdot \sigma_P}, \tag{7}$$

where $\text{Cov}(\cdot)$ denotes the covariance and $\sigma$ the standard deviation of the respective distributions.

- **Spearman Rank Correlation:** It assesses the monotonic relationship between the predicted and true rankings of probabilities, defined as,

$$\text{Spearman} = 1 - \frac{6 \sum_{i=1}^{|\mathcal{C}|} d_i^2}{|\mathcal{C}|(|\mathcal{C}|^2 - 1)}, \tag{8}$$

where $d_i$ is the difference between the ranks of $P_i$ and $\hat{P}_i$. Higher values indicate better ordinal agreement.

**Implementation Details.** `SpheREx` and all baselines are implemented using PyTorch. We fine-tune all parameters of the latent projector model BERT for all baselines and `SpheREx`except PMF. All training and inference tasks were conducted on an 48GB NVIDIA A6000 GPU to ensure high computational efficiency. Hyperparameters are discussed in Table 7.

## D QUORA QUESTION PAIR DATASET

**Data Preprocessing.** We begin by filtering out samples containing null or malformed questions from the Quora Question Pairs (QQP) dataset. A uniformly random subset comprising one-third of the cleaned dataset is selected, followed by an 80:20 stratified split into training and test sets, resulting in 107,827 training and 26,959 test instances. Each instance is formatted as a triplet $(q_a, q_b, \text{sim})$, where $q_a$ and $q_b$ denote the two questions and $\text{sim} \in \{0, 1\}$ indicates their semantic similarity. The textual input for each question is tokenized and processed using a pre-trained BERT encoder, with the `[CLS]` token representation extracted to serve as the input embedding for the `SpheREx` projection module.

**Model Training.** For each training instance $(q_a, q_b, \text{sim})$, we apply three complementary objectives: the geometric loss $L_g$, the probabilistic loss $L_p$, and the radius regularization loss $L_r$, as described in Section 3.4. To enforce the symmetry inherent in the semantic similarity relation, each sample is augmented with its reversed counterpart $(q_b, q_a, \text{sim})$. If $\text{sim} = 1$, we encourage strong semantic alignment by minimizing the containment-based overlap loss between the hyperspherical embeddings of $q_a$ and $q_b$. Conversely, when $\text{sim} = 0$, we apply a disjointness constraint that penalizes overlap between non-similar question embeddings. The final training objective is expressed as a weighted combination of the form:

$$L = \alpha L_g + \gamma L_r + \delta L_p,$$

where $\alpha$, $\delta$, and $\gamma$ are the respective weighting coefficients.

Table 8: Hyperparameters for experiments on Quora

| Name | Value | Significance |
|------|-------|-------------|
| n | 3 | number of layers in hypersphere projector |
| hidden | 64 | Dimension of hidden layer in hypersphere projector |
| embed_size | 6 | Dimensionality of the embedding space |
| batch_size | 512 | Batch size |
| lr | $2 \times 10^{-5}$ | Learning rate |
| $\alpha$ | 1.0 | Weight of geometric loss |
| $\gamma$ | 1.0 | Weight of regularization loss |
| $\delta$ | 1.0 | Weight of probabilistic loss |
| $\phi$ | 0.05 | Minimum radius of hyperspheres |
| $\tau$ | 0.5 | Threshold for assigning classes |
| epochs | 30 | Number of training epochs |
| Optimizer | AdamW | Optimizer used |
| Seeds | 42, 97, 137 | Seeds used |

**Inference.** To determine the semantic similarity between two query embeddings $(q_1, q_2)$, we compute a soft matching score $f_{\text{score}}(q_1, q_2)$ based on the degree of geometric overlap between their hyperspherical representations. Specifically, we define,

$$f_{\text{score}}(q_1, q_2) = 1 - \frac{\|\mathbf{c}_1 - \mathbf{c}_2\|_2}{r_1 + r_2},$$

where $\mathbf{c}_i$ and $r_i$ denote the center and radius of the hypersphere corresponding to $q_i$, for $i = 1, 2$. For the box-based baselines, we have considered,

$$f_{\text{score}}(q_1, q_2) = \min_{j=1,..n} \left( 1 - \frac{\mathbf{c}_{j1} - \mathbf{c}_{j2}}{\mathbf{r}_{j1} + \mathbf{r}_{j2}} \right)$$

where $n$ represents the number of dimensions, and $\mathbf{r}_j$ refers to side length in the particular dimension. The score captures the normalized proximity of the hyperspheres' centers, scaled by their total spread. A pair is classified as semantically similar ($sim_{\text{pred}} = 1$) if $f_{\text{score}}(q_1, q_2) > \tau$, and dissimilar ($sim_{\text{pred}} = 0$) otherwise, where $\tau$ is a predefined decision threshold. This formulation ensures that higher scores correspond to greater overlap and thus higher semantic affinity.

**Evaluation Metrics.** To assess the performance of SpheREx and baseline models on the QQP dataset, we employ standard classification metrics – Precision, Recall, and F1-score. Given a pair of questions $(q_1, q_2)$ and the corresponding ground-truth similarity label $y \in \{0, 1\}$, we compute the similarity score $f_{\text{score}}(q_1, q_2)$ as a function of the geometric overlap between their hyperspherical embeddings. A binary prediction $\hat{y}$ is assigned by thresholding this score with a tunable parameter $\tau$, i.e., $\hat{y} = \mathbb{I}[f_{\text{score}}(q_1, q_2) > \tau]$. We evaluate predictions over the entire test set and report:

- **Precision:** The proportion of predicted similar pairs that are truly similar,

$$\text{Precision} = \frac{\text{TP}}{\text{TP} + \text{FP}},$$

  where TP and FP denote the number of true positives and false positives, respectively.

- **Recall:** The proportion of truly similar pairs correctly identified by the model,

$$\text{Recall} = \frac{\text{TP}}{\text{TP} + \text{FN}},$$

  where FN is the number of false negatives.

- **F1-score:** The harmonic mean of precision and recall, providing a balanced measure of classification performance,

$$\text{F1} = 2 \cdot \frac{\text{Precision} \cdot \text{Recall}}{\text{Precision} + \text{Recall}}.$$

We also study model performance under varying threshold values $\tau$ to assess robustness in semantic similarity estimation. Final metrics are averaged across three random seeds for statistical reliability.

Table 9: Hyperparameters for experiments on Caltech-UCSD Birds-200-2011 Dataset

| Name | Value | Significance |
|------|-------|-------------|
| n | 3 | number of layers in hypersphere projector |
| hidden | 64 | Dimension of hidden layer in hypersphere projector |
| embed_size | 6 | Dimensionality of the embedding space |
| batch_size | 32 | Batch size |
| lr | $1 \times 10^{-7}$ | Learning rate |
| $\alpha$ | 1.0 | Weight of geometric disjoint loss |
| $\gamma$ | 1.0 | Weight of regularization loss |
| $\delta$ | 1.0 | Weight of probabilistic loss |
| $\phi$ | 0.05 | Minimum radius of hyperspheres |
| epochs | 60 | Number of training epochs |
| Optimizer | AdamW | Optimizer used |
| Seeds | 42, 97, 137 | Seeds used |

**Implementation Details.** SpheREx and all baselines are implemented using PyTorch. We fine-tune all parameters of the latent projector model BERT for all baselines and SpheREx. All training and inference tasks were conducted on an 48GB NVIDIA A6000 GPU to ensure high computational efficiency. Hyperparameters are discussed in Table 8.

# E CALTECH-UCSD BIRDS-200-2011 DATASET

**Data Preprocessing.** The CUB-200-2011 dataset consists of 11,987 bird images spanning 200 fine-grained species categories. We randomly select a subset of 20 classes and apply an 80:20 train-test split within this subset. Each image is preprocessed using the standard input pipeline from the open_clip library and encoded using the CLIP ViT-H/14 model pretrained on the LAION-2B dataset to obtain a dense image representation.

For each class label $l \in \mathcal{L}$, we construct a text prompt in the form "A label of ¡class name¿" and encode it using the same CLIP model to obtain the corresponding text embedding. We employ two separate SpheREx projection heads: one for image embeddings and another for text embeddings, ensuring modality-specific hyperspherical representations.

During training, for each image instance $i$, we sample its corresponding positive label $l_p$ and uniformly select a negative label $l_n \neq l_p$ from the remaining class set. Each training triplet $(i, l_p, l_n)$ is used to optimize the model with respect to both containment and disjointness constraints, encouraging the hyperspherical embedding of the image to lie within that of the correct label while maintaining separation from unrelated labels.

**Model Training.** Given each training triplet $(i, l_p, l_n)$, where $i$ denotes the image embedding, $l_p$ the correct label (positive), and $l_n$ a randomly sampled incorrect label (negative), the training objective optimizes both probabilistic and geometric consistency. Specifically, we enforce that the probabilistic containment score satisfies $P(l_p \mid i) \approx 1$ and $P(l_n \mid i) \approx 0$, where $P(l \mid i)$ is computed as the normalized volume of intersection between the hyperspherical embedding of $i$ and $l$.

In parallel, we enforce geometric disjointness between the hyperspheres of $l_p$ and $l_n$ using a margin-based separation constraint, preventing semantic collapse between unrelated classes. To avoid degenerate solutions (e.g., vanishing radii), we introduce a radius regularization term for each of the three hyperspheres $(i, l_p, l_n)$ to ensure their radii remain above a fixed minimum threshold. The final loss is composed of a weighted combination of probabilistic alignment, geometric separation, and radius regularization components.

**Inference.** During inference, given a test image embedding $i_{\text{test}}$, we compute a probabilistic matching score $f_{\text{score}}(i_{\text{test}}, l)$ for each candidate label $l \in \mathcal{L}$. The scoring function is derived from the normalized volume of intersection between the hyperspherical embeddings of the image and the label, effectively capturing the semantic compatibility between modalities. We then rank all labels based

on their respective scores and assign the label with the highest score as the predicted class for $i_{\text{test}}$, i.e., $l^* = \arg\max_{l \in \mathcal{L}} f_{\text{score}}(i_{\text{test}}, l)$.

**Evaluation.** We formulate the image-to-label prediction task as a ranking problem and evaluate model performance using three standard metrics: **Accuracy** (Hit@1), which measures the proportion of test images where the top-ranked label exactly matches the ground-truth; **Mean Rank (MR)**, which computes the average position of the correct label in the ranked list; and **Mean Reciprocal Rank (MRR)**, which calculates the average inverse rank of the correct label, thereby emphasizing early correct retrievals. These metrics collectively assess both exact match performance and the overall quality of the predicted ranking.

**Implementation Details.** SpheREx and all baselines are implemented using PyTorch. We fine-tune all parameters of the latent projector model CLIP for all baselines and SpheREx. All training and inference tasks were conducted on an 48GB NVIDIA A6000 GPU to ensure high computational efficiency. Hyperparameters are discussed in Table 9.

