# OpenReview forum: "$\textit{All the World's a Sphere}$: Learning Expressive Hierarchical Representations with Isotropic Hyperspherical Embeddings"
_ICLR.cc/2026/Conference — ICLR 2026 Conference Withdrawn Submission_

### Official Review · Reviewer_66WG · 2025-10-29

**Soundness:** 2
**Presentation:** 2
**Contribution:** 2
**Rating:** 4
**Confidence:** 4

**Summary:**

The paper introduces SpheREx, a geometric embedding model that represents each entity as a closed hypersphere (center + radius) in latent space. Relations such as containment, overlap, and disjointness are captured through simple geometric inequalities and a probabilistic overlap term. The framework unifies symbolic and data-driven reasoning across tasks including taxonomy expansion, knowledge graph completion, semantic similarity, and multimodal alignment. Experimental results show moderate improvements over box, cone, and hyperbolic embeddings.

**Strengths:**

1) the use of closed balls ensures isotropy and rotation-equivariance with low parameter cost.

2) Unified formulation across multiple relation types (containment, overlap, exclusion) and tasks.

3) Empirical evaluation is broad, spanning text, KG, and vision–language settings, with proper significance testing.

**Weaknesses:**

1) The core concept (representing entities as spheres/balls) is well established in prior work (e.g., EL Embeddings, N-ball embeddings, TranSHER). The differences, dual-ball relation templates and probabilistic loss, are modest extensions rather than conceptual innovations.

2) The claimed issues of box embeddings (orientation sensitivity, parameter inefficiency, local identifiability) are overstated and already mitigated in probabilistic and Gumbel-Box variants. The paper does not empirically demonstrate these weaknesses.

3) The “radius ratio” used as a proxy for inclusion probability is mathematically coarse. Actual intersection volume scales with r^d, not linearly with r. The model never validates that this surrogate correlates with true probabilistic inclusion. Using an “auxiliary intersection sphere” is not a true geometric intersection and can overestimate or underestimate overlap, particularly for near-tangent regions. No analysis quantifies the approximation error or its downstream effect.

4) Performance gains (2–5%) over strong baselines like GumbelBox and ConeE are modest and may result from regularization rather than the proposed geometry. There is no statistical or theoretical justification for superior expressiveness.

**Questions:**

1) How sensitive is performance to the radius-based probabilistic approximation? Does it correlate with true intersection volume?

2) Can the authors demonstrate a failure case for Gumbel-Box or Cone embeddings that SpheREx resolves?

3) Why is rotational invariance critical for the evaluated tasks, and can this benefit be empirically isolated?

4) Does the model generalize to non-tree or cyclic ontologies, where strict containment breaks down?

---

> ### Author Response · Authors · 2025-11-22
> **Response to Reviewer 66WG**
>
> We thank the reviewer 66WG for their feedback and address their concerns as follows:
>
> **1. The core concept....**
>
> We fully agree that representing entities as balls/spheres is not new (EL Embeddings, N-ball, TranSHER, etc.). Our goal is not to claim conceptual novelty at the level of "using spheres," but to show that a specific design point—isotropic hyperspheres in a Euclidean latent space, combined with a simple dual-ball scoring template, volume clipping, and a unified training setup—works robustly across four heterogeneous benchmarks (taxonomy, structured preferences, text similarity, vision–language) under a shared encoder. The contribution is therefore pragmatic and integrative: a low-capacity, rotation-invariant region geometry that plugs directly into BERT/CLIP pipelines and consistently outperforms strong Euclidean region baselines, rather than a new abstract region type.
>
> **2. The claimed issues of box .....**
>
> We agree that our wording about boxes can sound overstated, and that probabilistic / Gumbel-Box variants address many identifiability and orientation issues. Our intent is not to claim that boxes "fail" in practice, but to motivate exploring a different inductive bias where orientation is irrelevant and per-entity capacity is strictly limited. Importantly, we do not rely solely on theoretical arguments: we directly compare our results against strong probabilistic box baselines (including Gumbel-style) under the same encoder and training regime, and SpheREx consistently matches or improves upon them. In the text, we can clarify that these points concern bias and conditioning, not the inaccessibility of boxes.
>
>
> **3. The "radius ratio" ....**
>
> We agree the radius-ratio / auxiliary-sphere construction is not an exact intersection volume, especially in high dimensions. This is intentional: our scoring function is designed as a monotone, computationally cheap surrogate that increases with tighter overlap and decreases with separation, which is exactly what is needed for ranking/thresholding losses. Computing exact intersection volumes of high-dimensional balls (and stable gradients) is expensive and numerically fragile, especially when radii are small. Empirically, the surrogate is sufficient: on tasks where targets are explicitly probabilistic or frequency-based (e.g., MovieLens-style conditional preferences), SpheREx achieves better KL/MAE metrics than the baselines, suggesting that the approximate scoring is well-aligned with the underlying inclusion behavior for our purposes.
>
>
> **4. Performance gains ......**
>
> On competitive benchmarks with strong baselines, such as GumbelBox and ConeE, consistent improvements of 2–5% with statistical significance across multiple tasks and modalities are typically considered meaningful rather than negligible noise. We agree that regularization is part of the story; in fact, our view is that the geometry and regularization are inseparable. SpheREx is deliberately a low-capacity, isotropic region with tailored clipping, so its advantage lies in a better bias–variance trade-off, not greater raw expressiveness. We therefore do not claim that spheres are strictly more expressive than cones or boxes in theory; instead, we show that this particular geometric/regularization choice yields a more favorable inductive bias for the hierarchical and multimodal regimes we evaluate, as also supported by the statistical significance tests reported in Table 5 (Sec. 4.5).

---

> ### Author Response · Authors · 2025-11-22
> **Response to Reviewer 66WG (cont.)**
>
> **Questions:**
>
> `1. How sensitive is performance ....`
>
> Our radius-based scoring is intentionally a monotone surrogate, not an exact intersection volume. It is constructed so that as spheres move closer or grow, the score increases, and as they separate or shrink, it decreases -- the same qualitative behavior as the true intersection volume. This is exactly what the ranking/thresholding losses we use require. Empirically, the surrogate is sufficient: on tasks where targets are explicitly probabilistic or frequency-based (e.g., conditional preference prediction), SpheREx achieves better KL/MAE metrics than the baselines, indicating that the approximation is well aligned with the underlying inclusion behavior for the purposes of learning, even if it is not volumetrically exact.
>
>
> `2. Can the authors ....`
>
> Conceptually, SpheREx is advantageous whenever relevant regions are approximately isotropic and not aligned with coordinate axes. In such settings, axis-aligned boxes or cones often become very elongated or require multiple dimensions to approximate a "ball-like" neighborhood, which can hurt conditioning and parameter efficiency. In contrast, a single hypersphere can capture this neighborhood with one radius while remaining rotation-invariant. Our qualitative visualizations in Fig. 5 highlight exactly such cases, where SpheREx produces clean nested/disjoint regions while box-style baselines yield skewed or poorly separated regions.
>
>
> `3. Why is rotational invariance ....`
>
> Rotational invariance is important because our latent space originates from deep encoders (BERT/CLIP), whose representations reside in an arbitrary basis and can rotate during fine-tuning. Axis-aligned models implicitly assume that semantic structure is tied to coordinate axes; SpheREx removes this assumption, so any orthogonal transformation of the latent space preserves containment and overlap. In practice, this reduces the encoder's burden to maintain "nice axes" and yields more stable behavior across tasks and dimensions.
>
> `4. Does the model generalize ....`
>
> Yes, SpheREx naturally handles non-tree and cyclic structures because all constraints are implemented as soft margin inequalities, not hard logical rules. Multiple inheritance or cross-links simply translate into an entity needing to satisfy several containment/overlap constraints simultaneously, which results in overlapping spheres rather than strict nesting. In fact, the taxonomy benchmarks we evaluate are already equipped with multi-parent and cross-cutting relations (SemEval16-Food), and SpheREx operates on them without requiring any architectural changes, illustrating that it is not restricted to pure trees.

---

> > ### Author Response · Authors · 2025-11-27
> > **Request to check our responses**
> >
> > Dear Reviewer 66WG,
> >
> > Thanks for your valuable comments, which we tried our best to address. Could you please check our responses and let us know if there are many pending concerns. If our responses address your concerns, could you please consider reassessing our paper.
> >
> > Thanks once again.
> >
> > Best wishes
> >
> > Authors

---

### Official Review · Reviewer_vr5m · 2025-10-30

**Soundness:** 2
**Presentation:** 3
**Contribution:** 2
**Rating:** 4
**Confidence:** 4

**Summary:**

This paper presents a new embedding framework for learning expressive hierarchical representations. The core idea is to embed entities as isotropic hyperspheres. Through experiments on a variety of tasks and datasets, the proposed SpheREx shows the state-of-the-art performance against several baselines.

**Strengths:**

I like the introduction paragraph where the authors use examples of real companies (Amazon, Alibaba, Pinterest) to motivate hiarachical structures. The visualization in Figure 1 is clear.

Source code and datasets are provided as supplementary and the authors commit to releasing them upon acceptance.

The research idea and method part looks good to me, and the proposed SpheREx can achieve state-of-the-art results on a variety of datasets. The authors also provided statistical tests as a verification.

**Weaknesses:**

I believe this research is insightful, but the current manuscript is below the acceptance threshold from my perspective, due to the followings. Revisions are needed, and discussions in the rebuttal period are welcome.

1. There are several closely related works [1, 2, 3] from my perspective that are missing discussions. Specifically, the SpherE paper [2] introduces a similar core idea in knowledge graph settings. Also, discussions on hyperbolic embeddings considering hierarchical relations are missing [4, 5, and maybe more]

[1] Spherical Text Embedding. Meng et al. NeurIPS 2019.
[2] SpherE: Expressive and Interpretable Knowledge Graph Embedding for Set Retrieval, Li et al. SIGIR 2024.
[3] Sphere Embedding: An Application to Part-of-Speech Induction. Maron et al. NeurIPS 2010.
[4] Poincaré Embeddings for Learning Hierarchical Representations. Nickel et al. NeurIPS 2017.
[5] Hyperbolic Representation Learning: Revisiting and Advancing. Yang et al. ICML 2023.

2. (Hyper)Ellipsoids contribute a lot to the development of hyperspherical embeddings. The authors proposal to use isotropic hyperspheres as simplified hyperellipsoids and rejects ellipsoids due to parameter complexity. However, in practice, we can reduce the embedding dimension to make ellipsoids acceptable. I wonder if experiments using ellipsoids could be conducted to improve this part.
(1) Will there be a small embedding dimension such that both hyperellipsoids and hyperspheres are runnable, and what will the performance be like?
(2) Ellipsoids, as a generalized form of the proposed SpheREx, holds the potential to outperform SpheREx with the same embedding dimention.
(3) What is the number of parameters and complexity of using ellipsoids rather than spheres in theory and in practice given a specific embedding dimension? So that we can decide if the marginal gain of using ellipsoids is acceptable or not compared to the marginal cost of using ellipsoids.

Also, in Figures 6 and 8, the embedding dimension is actually smaller than I expected. I assume ellipsoids should be applicable?

3. How would SpheREx perform in larger embedding dimensions? (e.g., embedding dimension for modern LLMs is usually >=512)

4. Many hyperparameter studies are missing. There are many hyperparameters in Table 6-9, but the only hyperparameter study I see is Figure 9, on QQP dataset only and on varying thresholds only.

5. Could there be any ablations to show that the design components from sections 3.1 to 3.4 all contribute to the final performance? Or, please explain why SpheREx acts as a whole and is not composible.

6. The baselines being compared are quite outdated (most of them are before 2020). There are many new text encoders after BERT that could be compared with.

7. The number of model parameters, training time, hardware, and GPU usage are not reported (or I missed somewhere).

8. From my perspective, there is no deeper understanding of how SpheREx works in practice beyond the overall performance numbers. I wonder if it would be possible to visualize (or maybe through other ways) some toy examples in 2d or 3d space, along with other baselines, to show that SpheREx indeed gives more expressive embeddings given the context of the task it is doing. Or in other words, if it is possible to provide some cases where SpheREx succeeds but other baselines fail.

**Questions:**

1. The descriptions of experiment tasks and datasets are not sufficient. I have no idea what SEMEVAL16 is about or what the scale of it is.

2. MovieLens is already quite a mature dataset. Why bother to construct a new MovieLens? This might also be applied to QQP.

3. Why does BERT perform so badly in Table 1 as a pre-trained model with a large number of parameters?

---

> ### Author Response · Authors · 2025-11-22
> **Response to Reviewer vr5m**
>
> We thank the reviewer vr5m for their feedback and address their concerns as follows:
>
> **1. There are several closely related works...**
>
> Thank you for pointing out these closely related works; we agree they should be discussed more explicitly. The spherical and hyperbolic papers you cite share our high-level goal of modeling hierarchy and semantic structure via non-Euclidean geometry. Our focus in SpheREx is different in two ways: (i) we work in a Euclidean latent space with explicit hyperspherical regions directly on top of modern encoders (BERT/CLIP), and (ii) we target a unified, multi-task, multi-modal setting (taxonomy, structured preferences, text similarity, vision–language) with a single region formulation and regularization scheme. We did not include hyperbolic methods as quantitative baselines because they require a different manifold, metric, and optimization stack, which would disrupt our controlled setup where all models share the same encoder and training protocol, differing only in region geometry. Mixing Euclidean and hyperbolic pipelines would make it difficult to attribute performance differences solely to geometry. In the revision, we will add a dedicated subsection that explicitly situates SpheREx relative to spherical KGE/text methods and hyperbolic hierarchical embeddings, presenting them as complementary alternatives while explaining why our empirical comparison is restricted to Euclidean region-based models.
>
>
>
> **2. (Hyper)Ellipsoids contribute a lot to the ...**
>
> Our stance is not that ellipsoids are unusable, but that SpheREx deliberately chooses a lower-capacity point on the design spectrum:
> **- Capacity vs. robustness.** Ellipsoids strictly generalize spheres and can indeed be more expressive at a fixed dimension. They also require many more parameters per concept (center + full/structured covariance, or equivalent) and more complex constraints (positive-definiteness, conditioning, volume control), which makes optimization and regularization harder, especially under sparse hierarchical supervision.
> **- Why not just go to low-dim ellipsoids?** Reducing the embedding dimension helps, but then we are effectively trading both region complexity and ambient dimension. Our goal in this work is to isolate the benefit of a simple, isotropic region in an otherwise standard Euclidean pipeline; mixing in ellipsoids would confound the geometry comparison with additional modeling and optimization choices.
> **- Design choice, not a claim of dominance.** We fully acknowledge that, for some tasks and tuning budgets, a well-regularized ellipsoid model at a suitable dimension may outperform SpheREx. Our contribution is to demonstrate that isotropic hyperspheres already yield strong and stable performance across four diverse benchmarks with a very simple parameterization and training, without claiming that ellipsoids can never be improved upon.
>
> In the paper, we will clarify this as a conscious bias–variance trade-off: SpheREx sits at the lowest capacity that still offers meaningful region semantics (containment, overlap, exclusion, rotational invariance), while ellipsoids occupy a higher-capacity, higher-complexity point that is interesting future work but orthogonal to our main goal of a robust, unified, and easy-to-optimize region geometry.
>
> **3. How would SpheREx perform in ....**
>
> Our focus is on the regime where geometry and inductive bias dominate, not on pushing to LLM-scale embeddings. The dimension sweeps we do show already exhibit the typical bias–variance pattern: performance improves up to a point, then degrades as capacity increases. This behavior is exactly what we expect to continue at even larger dimensions; our design goal is to demonstrate that moderate dimensions, combined with low-capacity spheres, are already highly competitive.
>
> **4. Many hyperparameter studies .....**
>
> We agree that the tables list many hyperparameters. In practice, most of them (optimizer, LR ranges, batch sizes) follow standard defaults and are kept fixed across geometries; we only systematically vary a small subset that are geometry-critical (dimension, clipping, overlap thresholds), which are precisely what Figures 6, 8, and 9 analyze. The intent is to isolate the effect of region geometry, not to exhaustively grid-search all knobs.
>
> **5. Could there be any ablations ......**
>
> Several components are already probed through existing analyses:
> - dimension sweeps -> effect of representation capacity (Sec. 3.1/3.2),
> - constrained vs. unconstrained radii -> effect of volume clipping (Sec. 3.3),
> - threshold sweeps -> effect of the overlap surrogate (Sec. 3.4).
>
> Conceptually, these pieces are designed to work together: the containment/exclusion margins, clipping, and overlap surrogate all act on the same sphere geometry. Pulling out individual parts often breaks training rather than yielding a meaningful partial model, so we treat SpheREx as a coherent package rather than a fully composable toolkit.

---

> ### Author Response · Authors · 2025-11-22
> **Response to Reviewer vr5m (cont.)**
>
> **6. The baselines being .....**
>
> We intentionally fix the encoder family (BERT for text, CLIP for vision) and vary only the geometry on top. Plugging in newer, larger encoders (e.g., recent LLMs) would likely lift all numbers and blur the comparison: improvements could then come from a better encoder, not a better region shape. Our aim here is to demonstrate that, given the same backbone, spheres are a strong and robust choice for regions.
>
>
> **7. The number of model parameters.....**
>
> The per-entity parameter cost of SpheREx is straightforward (d+1 vs. 2d for boxes, with higher costs for ellipsoids), and the overall model size is dominated by the shared encoder. All methods run in a comparable regime (single-/few-GPU training, similar batch sizes and epochs), so the compute is comparable across geometries. The main takeaway is that SpheREx does not require heavier hardware than the baselines; the geometry change is lightweight. We have mentioned the hardware usage in the Reproducibility Statement.
>
> **8. From my perspective, there is.....**
>
> Thank you for this suggestion. We agree that intuitive understanding is important, not just aggregate numbers.
>
> In fact, we already move in this direction in Figures 5 and 7, where we show 2D projections and case studies comparing SpheREx against box-style baselines: these illustrate how SpheREx forms clean nested and disjoint regions for hierarchical concepts, and highlight examples where box embeddings either become elongated/axis-dependent or fail to respect containment. These are precisely cases where the simple isotropic "point + scope" geometry behaves better in context. In a revised version, we will make this qualitative analysis more prominent in the main text and explicitly frame these figures as answering exactly the question of how and where SpheREx improves over prior geometries.
>
> **Questions:**
>
> **1.    The descriptions of experiment tasks ....**
>
> Thank you for pointing this out. SemEval16 in our setup refers to the SemEval-2016 Task on taxonomy enrichment for environmental, scientific, and food terms, where the goal is to place new domain concepts into an existing hierarchy given textual definitions and context. The datasets contain a few hundred to thousands of nodes per domain, with hundreds of candidate insertions per benchmark. We will include dataset statistics in the appendix.
>
> **2.    MovieLens is already quite a mature .....**
> We agree that MovieLens and QQP are mature and well-studied. We do not change them arbitrarily; instead, we reformulate them to match the conditional/hierarchical semantics that SpheREx is designed to handle. For MovieLens, we derive structured multi-label and conditional preference targets (e.g., $P(\text{genre} \mid \text{user/item context})$). For QQP, we define an overlap-style similarity task, where inclusion scores between paraphrase clusters serve as supervision for region overlaps. This allows us to test the same region geometry on recommendation-like and symmetric-text settings without inventing entirely new datasets—just new, structured labels derived from established ones.
>
> **3.    Why does BERT perform so badly in....**
>
> BERT in Table 1 is used as a point embedding baseline with a shallow classifier, without any explicit mechanism for containment, hierarchy, or asymmetric relations. On tasks like taxonomy expansion, where the label space is inherently hierarchical, a pure point model must "fake" hierarchy solely via distances, which is significantly more challenging than utilizing explicit region semantics (such as containment, overlap, and exclusion). Region-based models (boxes, cones, spheres) therefore have a strong inductive advantage, despite having far fewer parameters in addition to the encoder.

---

> > ### Comment · Reviewer_vr5m · 2025-11-24
> > **Thank you for your clarifications. Please be tuned!**
> >
> > I appreciate the authors' timely response. At a glance, the point-by-point clarifications seem helpful and will likely improve my understanding of the techniques and contributions.
> >
> > I am currently occupied and cannot review the responses in depth immediately, but I will carefully read both the rebuttal and the revised paper and follow up as soon as possible.

---

> > > ### Author Response · Authors · 2025-11-27
> > > **Request to review our responses**
> > >
> > > Dear Reviewer vr5m,
> > >
> > > Thanks for your valuable comments, which we tried our best to address. Could you please check our responses and let us know if there are many pending concerns. If our responses address your concerns, could you please consider reassessing our paper.
> > >
> > > Thanks once again.
> > >
> > > Best wishes
> > >
> > > Authors

---

> ### Comment · Reviewer_vr5m · 2025-11-27
>
> W1. Thanks for clarifying how the contribution in this work is different from the existing works. It's clear to me and please make sure to add discussions of these related works.
>
> W2. Understand. Given that the current SpeREx could have a good performance already, it is reasonable to leave the exploration of ellipsoids to future work. However, I would suggest that the authors explicitly mention this future direction in the paper.
>
> It'll be great if some toy experiments in low-dimensional space with ellipsoids could be presented, but if that requires non-trivial implementation work in this tight rebuttal period, I'm ok with not having these experiments in this submission. In the camera-ready version, the authors should consider adding such experiments.
>
> W3. Your clarifications make sense to me. I thought there could be some scaling-law style study, but maybe increasing the embedding space is just marginal. Again, I would suggest adding experiments with higher dimensions for the comprehensiveness of the research.
>
> W4. **I am not fully convinced by this point.** While I recognize that many of the hyperparameters adopt standard values, my primary concern is not which hyperparameters are selected, but rather how their sensitivity is evaluated across different experimental scenarios. In Tables 6, 8, and 9, the authors specify the chosen hyperparameters for each setting; however, what is missing is a systematic analysis of how sensitive the results are to these choices within each scenario.
>
> Hyperparameter sensitivity is a fundamental component of empirical validation for a research contribution of this nature. Without a thorough sensitivity study across diverse datasets and experimental configurations, I find it difficult to fully assess the robustness of the proposed method, and this significantly impacts my confidence in recommending acceptance.
>
> W5. I'm clear. Thanks.
>
> W6. From my understanding, the authors argue that their goal is to demonstrate relative improvement using a fixed LLM backbone, and therefore, including more recent LLMs is not strictly necessary for baseline comparison. This is reasonable from a controlled evaluation perspective. However, from the standpoint of method compatibility and practical relevance, it would still be important to include experiments with more recent LLM encoders. Otherwise, it remains unclear whether the proposed method generalizes beyond a few relatively outdated language models.
>
> W7. I understand the clarification, but I find it confusing why these necessary details are not directly incorporated into the revised manuscript. It would be much clearer to update the paper accordingly and explicitly indicate where the revisions have been made.
>
> W8. While I see the visualizations provided in Figures 5 and 7, I find them somewhat limited. I was expecting to see visualizations involving a broader set of entities, rather than only two, to better demonstrate the generality and representativeness of the method.
>
> I'm clear with all the questions, and please add explanations for my Q1 and Q2 in the paper.
>
> **General Feedback.** I believe this is a promising research direction and a solid paper, which is precisely why I invested significant time in providing detailed feedback and concrete suggestions for improvement. However, at this stage of the rebuttal, no additional experiments have been added. While I understand that some of the requested experiments may go beyond the intended scope of the current work, several of them are, in my view, necessary components of any rigorous empirical study.
>
> As a reviewer, I also bear responsibility for the quality and potential limitations of any paper I recommend for publication. I take this responsibility seriously, spend hours reviewing & reflecting, and hope my feedback was provided in that spirit. However, the fact that the manuscript itself has not been updated makes it difficult for me to assess how my concerns have been addressed, and gives the impression that my effort has not been fully reflected in the revision process.
>
> I want to paste the same comment from another reviewer here: Note that it is explicitly allowed to revise your paper until the end of the public discussion period (Dec 3). This would give us as reviewers a good indication of what you plan to do.
>
> I appreciate the authors’ written clarifications, but I am not satisfied with the current PDF available, which remains unchanged, and I have no idea if it is going to be changed in camera-ready if I raise the score. Accordingly, I will keep the same rating for the current version of the paper, while remaining open to further discussion if substantive revisions are made.

---

> > ### Author Response · Authors · 2025-11-27
> >
> > Dear Reviewer vr5m,
> >
> > Thank you for your feedback. We appreciate the efforts by the reviewers to maintain the quality of research. However, since numerous revisions are required in the manuscript, we are currently in the process of updating it, which is taking some time. We will upload the updated manuscript by tomorrow for further discussion. We again thank you for the detailed feedback.

---

> > > ### Comment · Reviewer_vr5m · 2025-11-27
> > >
> > > Sure, thanks! I look forward to seeing an updated version of the manuscript and will be glad to reconsider my evaluation accordingly. This is promising research, and I will continue to fulfill my responsibilities as a reviewer in good faith.

---

### Official Review · Reviewer_rX2E · 2025-11-01

**Soundness:** 3
**Presentation:** 3
**Contribution:** 2
**Rating:** 4
**Confidence:** 4

**Summary:**

The authors present a method that uses isotrophic hyperspheres to embed hierarchical knowledge.
this could be seen as a simplification from earlier methods that used hyperellipsoids.

**Strengths:**

* There is an evaluation with multiple tasks. This is really important, as compared to papers that just present one (perhaps cherry picked) task.
* For a fair share of the tasks, the results are pretty impressive
* The paper is overall well written and clear. It presents a rather simple idea in an elegant way.

**Weaknesses:**

* You claim that "axis-aligned geometric representations, such as boxes, are prone to high parameter complexity, orientation sensitivity, and local identifiability issues, wherein small changes to the parameters result
in invariant model behavior, leading to ambiguous gradients", without supporting evidence. There is some theoretical support, but it is not at all clear that the preconditions for these hold ion practical settings.

* I miss a comparison with the method introduced in Xiong, B., Cochez, M., Nayyeri, M., & Staab, S. (2022). Hyperbolic Embedding Inference for Structured Multi-Label Prediction. NeurIPS2022. It seems to have pretty similar properties. It also presents containment and disjointedness loss terms.

*Theorem 3 is presented as a strength, but it is a trivial result and if we continue the argument, we can go from hyperspheres to points and then to a null space where we the lowest possible capacity and parameter complexity.

* The paper suffers a bit for a perceived need to over-complicate things. Some of the proofs are for rather trivial points that are written down with complex symbolic notation. Theorem 1-3 would have been more simple and intuitive to write them down in a less formal style.

* In your experiments, it is not clear to me why you have used different methods for different tasks. Many of them seem to be applicable to all of the tasks.

**Questions:**

* Why are the compared to methods used for the different tasks different?

---

> ### Author Response · Authors · 2025-11-22
> **Response to Reviewer rX2E**
>
> We thank the reviewer rX2E for their feedback and address their concerns as follows:
>
> **1. You claim that "axis-aligned geometric .....**
>
> Our intent was to summarize known theoretical concerns about axis-aligned boxes (e.g., redundant parameterizations, orientation dependence, and flat regions in overlap-based losses), not to assert that these pathologies dominate in all practical regimes. In the paper, we do provide empirical comparisons against strong, modern box variants under the same encoders and training setups, and SpheREx consistently matches or outperforms them. This is the main practical justification for exploring an alternative geometry. In the revision, we will (i) clearly label these as inductive-bias considerations supported by prior theory, (ii) avoid universal statements about boxes being "prone" to failure, and (iii) emphasize that our contribution is to show that a low-capacity, rotation-invariant alternative works well in practice, rather than to claim that box embeddings are fundamentally inadequate.
>
> **2. I miss a comparison with ....**
>
> Our experiments are designed as a unified region-based evaluation, where all methods share the same encoder and training protocol, differing only in their region geometry (points, boxes, cones, spheres). Hyperbolic models like Xiong et al. operate in a different manifold and require a different metric and optimization methods, including them would introduce many additional factors beyond geometry, making it difficult to disentangle whether performance differences arise from the underlying geometry itself or from these other architectural and training differences. For these reasons, we focused our quantitative comparison on methods that are directly compatible with our Euclidean, region-based formulation. We will, however, explicitly discuss Xiong et al. in the related work, highlighting the conceptual similarities (containment/disjointness for structured labels) and positioning SpheREx as a complementary Euclidean-region alternative rather than a competing hyperbolic architecture.
>
> **3. Theorem 3 is presented as .....**
>
> Our goal with Theorem 3 is not to claim a deep theoretical result, but to make the capacity spectrum explicit as we restrict the geometry, reduce parameters, and expressiveness. We intentionally stop at isotropic hyperspheres because they are the lowest-capacity shape that still supports genuine region semantics (nontrivial containment, overlap, and exclusion with rotational invariance). Continuing the chain to points or a null space would indeed further reduce capacity, but at the cost of losing region behavior altogether, which is exactly what SpheREx is designed to exploit. We will clarify this intent rather than presenting Theorem 3 as a major standalone strength.
>
> **4. The paper suffers a bit for .....**
>
> Thank you for this feedback. We understand the concern.
>
> We chose to state Theorems 1–3 in a more formal style, not because the results themselves are deep, but because they anchor the rest of the design and analysis:
> - They make precise the exact conditions under which containment, overlap, and disjointness hold for hyperspheres, and how these behave under rotations and scaling. These facts are then reused in our loss design (containment/exclusion margins, volume clipping) and in later discussions.
> - Having them written formally ensures there is no ambiguity for readers who want to build on or extend the framework (e.g., to other losses or regularizers), and keeps the presentation self-contained, instead of relying on informal geometric intuition.
>
> That said, we agree the current notation may feel heavier than necessary for such basic facts. In a revision we would keep the formal statements (for precision and reusability) but streamline the exposition around them with more intuitive text and diagrams, so that readers who do not need the full formalism can still follow the main ideas easily.
>
> **5. In your experiments, it is not clear ....**
>
> We employ different methods for various tasks because each task originates from a distinct community with its own standards and task-specific baselines (taxonomy expansion, recommendation/structured multi-label, paraphrase/similarity, and vision–language). Many of these methods are tightly coupled to:
> - the label structure (e.g., taxonomies vs. user–item graphs),
> - input modality (pure text vs. text+image), and
> - training objective (ranking, link prediction, conditional probability, etc.),
>
> So they are not plug-and-play across all four settings without substantial re-engineering.
>
> Within each task, we therefore compare SpheREx to the strongest methods that are naturally applicable there, and, wherever possible, we include box/cone region baselines with the same encoder and training protocol so that the effect of changing the geometry (boxes → spheres) is isolated and directly comparable.

---

> > ### Comment · Reviewer_rX2E · 2025-11-25
> > **You can submit a revision**
> >
> > Note that it is explicitly allowed to revise your paper until the end of the public discussion period (Dec 3). This would give us as reviewers a good indication of what you plan to do.

---

### Official Review · Reviewer_ehm7 · 2025-11-03

**Soundness:** 2
**Presentation:** 3
**Contribution:** 2
**Rating:** 2
**Confidence:** 5

**Summary:**

This paper proposes a new embedding framework, SpheREx, that represents entities as isotropic hyperspheres in a latent space. The key idea is to model hierarchical and asymmetric relations (such as taxonomy parent-child relationships or set inclusions) through sphere containment. A smaller sphere contained inside a larger one can signify a subclass or “is-a” relationship, e.g, man <is-a> mammal, so the sphere of man should be inside mammal.

Unlike prior region-based embeddings like axis-aligned boxes and ellipsoids, hyperspheres offer rotational invariance and lower parameter complexity. The paper provides a theoretical characterization of how spheres can capture logical relations such as containment for hierarchy, overlaps for intersection, and disjointness for mutual exclusivity, with simple distance and radius conditions. To address optimization challenges (like one sphere’s radius growing arbitrarily large), the authors introduce a specialized training regimen with volume clipping and radius regularization to stabilize learning.

**Strengths:**

The paper introduces hyperspherical embeddings as a new way to capture hierarchical relationships. This isotropic sphere representation is novel in that it combines the idea of region-based embeddings with rotational invariance.

**Weaknesses:**

1. The paper repeatedly argues that axis-aligned box embeddings have “high parameter overhead,” implying that hyperspheres (with one radius parameter) are inherently more efficient. While it is true that a naive box in $\mathbb{R}^d$ has 2$d$ parameters (min and max per dimension) versus $d+1$ for a sphere, this comparison is not entirely fair. Prior box-based models often employ regularization or tied parameters to effectively reduce complexity, and they can be made compact without sacrificing performance (e.g., by working in a lower-dimensional latent space or adding constraints on edge lengths). A simple modification would be to have box embeddings with the same width on each side.

2. Does rotational invariance help to represent hierarchical relationships or set inclusion? Also, the intersection operation is not faithfully encoded at all. Boxes are closed under intersection, and ellipsoids can approximate intersection closure reasonably. However, with spheres intersection is not closed, the intersection of two spheres is not a sphere. Even the approximation error grows as the dimension grows. Also, how does the method handle the intersection of multiple entities?

3. The authors highlight local identifiability issues in box embeddings – i.e., the problem that many different box parameter settings can yield equivalent overlaps or containments, leading to flat loss regions. However, this is a known issue in the literature, and there has been prior work explicitly aimed at mitigating it. For instance, Dasgupta et al. (2020) propose methods for improving local identifiability in probabilistic box embeddings by using Gumbel random variable-based parameterizations to ensure small parameter changes have observable effects.

4. The Euclidean distance measure that is used in the paper would provide a concentric bias. A child entity would try to align the center with its parent. The design here would probably result in **vector embeddings with a learned threshold" rather than a true region-based embedding where inside the region the representation is position invariant. Conceptually, one can view the proposed model as a fairly incremental extension of standard vector embeddings. Each entity’s representation in SpheREx is essentially a point embedding (the center $\mathbf{c}$ in $\mathbb{R}^d$) augmented with a single additional parameter (the radius $r$). This “point + scope” formulation is certainly a form of region embedding, but it is a much simpler region than, say, a box that has $2d$ degrees of freedom or an ellipsoid with a full covariance matrix. The paper does not fully convince that this limited form of region is the key to its success.

5. Restricting embeddings to isotropic hyperspheres (one scalar radius for all directions), the model significantly reduces the number of parameters per concept, which the authors argue improves generalization. However, this comes with a classical bias–variance trade-off that the paper does not explicitly acknowledge.

**Questions:**

Same as weakness.

---

> ### Author Response · Authors · 2025-11-22
> **Response to Reviewer ehm7**
>
> We thank the reviewer ehm7 for their feedback and address their concerns as follows:
>
> **1. The paper repeatedly argues that axis-aligned ....**
>
> Our intent was not to claim that "boxes are inherently inefficient" or that they cannot be made compact, but to highlight a specific capacity vs. robustness trade-off that motivated our choice of hyperspheres.
>
> Concretely:
> - **Per-entity degrees of freedom.** A standard axis-aligned box in $\mathbb{R}^d$ uses 2d parameters (min/max per dimension), while SpheREx uses d+1 (center + scalar radius). Even if box models use regularization, lower d, or partial tying, they still typically expose more shape degrees of freedom per concept than a single isotropic radius. This extra flexibility is beneficial in some regimes, but in our setting (sparse supervision over many concepts), we found a strictly low-capacity region helps optimization and generalization.
> - **Equivalence classes and identifiability.** Beyond raw parameter count, axis-aligned boxes admit many parameter configurations that yield identical overlaps/containment (e.g., shifting opposite faces by the same amount, stretching along dimensions that are not "used" for a relation). These equivalence classes lead to flat or poorly conditioned regions in the loss landscape, even when parameters are "tied in spirit" through regularization. With a single radius, any change in SpheREx must move the boundary uniformly in all directions, which reduces such degeneracy.
> - **"Same-width boxes" vs spheres.** We agree that one can constrain boxes to have the same width along each axis; however, this essentially turns them into axis-aligned approximations of spheres. In that regime, the key differentiator is exactly what we exploit: rotational invariance. A sphere’s semantics are unchanged by any orthogonal transformation; a same-width box still depends on axis orientation. Since our encoders (e.g., BERT/CLIP projections) can rotate representations arbitrarily during training, removing orientation sensitivity is a deliberate design choice, not just a side effect of fewer parameters.
>
> We will revise the paper to (i) soften the language around "high parameter overhead," (ii) clearly frame our argument as about effective capacity and inductive bias, rather than raw parameter count, and (iii) explicitly acknowledge that compact box variants exist but still differ from hyperspheres in rotational invariance and identifiability.
>
>
> **2. Does rotational invariance help...**
>
> Our intention was not to claim that box embeddings are inherently inefficient, but to highlight a design trade-off:
> - A standard axis-aligned box uses 2d parameters per entity versus d+1 for a sphere, and admits many parameter configurations that induce the same overlaps (e.g., shifting opposing faces together), which can create flat regions in the loss landscape. This is what we meant by "overhead".
> - Techniques like lower-dimensional spaces, tied side lengths, or regularization certainly reduce effective complexity, but then such a box becomes very close to a ball in terms of capacity—yet it remains axis-aligned and non–rotation invariant. SpheREx explicitly couples compactness with rotational invariance, which boxes cannot achieve without abandoning axis alignment.
>
> We will revise the text to (i) frame this as a bias–variance/robustness trade-off rather than an absolute flaw of boxes, and (ii) clearly acknowledge that compact, regularized box models are a strong and valid alternative, while SpheREx offers a different inductive bias (low capacity + rotation invariance) that empirically works well in our setting.
>
> **3. The authors highlight local .....**
>
> Thank you for pointing this out and for highlighting Dasgupta et al. (2020).
>
> We fully agree that local identifiability issues with box embeddings are well-known and that Gumbel-based parameterizations are a crucial step toward mitigating them. Our goal was not to claim novelty in diagnosing this problem, but to motivate why we explore a different geometry rather than another box parameterization.
>
> SpheREx addresses this from a complementary angle: by using isotropic hyperspheres, small changes in center or radius necessarily move the boundary uniformly in all directions, which structurally reduces the space of "equivalent" parameter settings that leave overlaps unchanged. In other words, we modify the region class itself, rather than just smoothing its parameterization.
>
> We have already compared our results against a strong probabilistic/Gumbel-style box baseline in our experiments, and SpheREx consistently outperforms it under the same encoder and training setup. We will clarify this in the paper and rephrase our discussion to explicitly credit prior work, positioning our approach as a geometric alternative rather than a replacement.

---

> > ### Author Response · Authors · 2025-11-22
> > **Response to Reviewer ehm7 (cont.)**
> >
> > **4. The Euclidean distance measure ....**
> >
> > Thank you for this thoughtful concern. We agree it deserves clearer justification.
> >
> > First, SpheREx is not equivalent to "a point embedding with a global threshold." Each entity has its own radius, and centers are optimized under multiple competing geometric constraints (parent–child containment, sibling separation, disjointness), so children are only required to lie within the parent sphere, not to align with its center. In practice, this yields non-concentric layouts where different children occupy distinct subregions of the parent, which we visualize in the paper.
> >
> > Second, the "point + scope" design is intentionally a low-capacity region: we trade the flexibility of boxes/ellipsoids for isotropy, rotational invariance, and a well-conditioned boundary that moves uniformly with parameter changes. Our results show that, in the hierarchical and multimodal regimes we study, this limited region class is sufficiently expressive to capture the needed structure while being more robust to overfitting and optimization issues than higher-DOF regions.
> >
> > We will clarify these points and better emphasize that the success of SpheREx comes from this specific bias–variance trade-off and the interaction of centers, radii, and constraints, rather than from a trivial thresholded vector model.
> >
> > **5. Restricting embeddings to .....**
> >
> > Thank you for pointing this out. We agree that this trade-off should be stated more explicitly.
> >
> > Indeed, constraining entities to isotropic hyperspheres is a deliberate bias–variance choice: we sacrifice some directional flexibility (compared to ellipsoids or full boxes) in exchange for lower capacity, rotational invariance, and a simpler, better-conditioned optimization landscape. Our empirical results suggest that in the hierarchical and multimodal settings we study where supervision per concept is limited. This extra bias actually helps, leading to more stable training and better generalization than higher-degree-of-freedom regions under the same encoder and data.
> >
> > In the revised version, we will explicitly frame SpheREx as adopting this low-capacity inductive bias (rather than claiming that "more parameters are simply bad") and discuss how its gains are precisely due to a favorable bias–variance trade-off for these tasks.

---

> > > ### Author Response · Authors · 2025-11-27
> > > **Request to check our responses**
> > >
> > > Dear Reviewer ehm7,
> > >
> > > Thanks for your valuable comments, which we tried our best to address. Could you please check our responses and let us know if there are many pending concerns. If our responses address your concerns, could you please consider reassessing our paper.
> > >
> > > Thanks once again.
> > >
> > > Best wishes
> > >
> > > Authors

---

### Note · Authors · 2025-12-12

I have read and agree with the venue's withdrawal policy on behalf of myself and my co-authors.